# OnePose++: Keypoint-Free One-Shot Object Pose Estimation without CAD Models

**Xingyi He**[1*]    **Jiaming Sun**[2*]    **Yuang Wang**[1]    **Di Huang**[3]
**Hujun Bao**[1]    **Xiaowei Zhou**[1†]

[1]Zhejiang University       [2]Image Derivative Inc.       [3]The University of Sydney

## Abstract

We propose a new method for object pose estimation without CAD models. The previous feature-matching-based method OnePose [48] has shown promising results under a one-shot setting which eliminates the need for CAD models or object-specific training. However, OnePose relies on detecting repeatable image keypoints and is thus prone to failure on low-textured objects. We propose a keypoint-free pose estimation pipeline to remove the need for repeatable keypoint detection. Built upon the detector-free feature matching method LoFTR [47], we devise a new keypoint-free SfM method to reconstruct a semi-dense point-cloud model for the object. Given a query image for object pose estimation, a 2D-3D matching network directly establishes 2D-3D correspondences between the query image and the reconstructed point-cloud model without first detecting keypoints in the image. Experiments show that the proposed pipeline outperforms existing one-shot CAD-model-free methods by a large margin and is comparable to CAD-model-based methods on LINEMOD even for low-textured objects. We also collect a new dataset composed of 80 sequences of 40 low-textured objects to facilitate future research on one-shot object pose estimation. The supplementary material, code and dataset are available on the project page: https://zju3dv.github.io/onepose_plus_plus/.

## 1   Introduction

Object pose estimation is crucial for immersive human-object interactions in augmented reality (AR). The AR scenario demands the pose estimation of arbitrary household objects in our daily lives. However, most existing methods [39, 29, 38, 55, 2, 4, 37] either rely on high-fidelity object CAD models or require training a separate network for each object category. The instance- or category-specific nature of these methods limits their applicability in real-world applications.

To alleviate the need for CAD models or category-specific training, OnePose [48] proposes a new setting of *one-shot object pose estimation*. It assumes that only a video sequence with annotated object poses is available for each object and aims for its pose estimation in arbitrary environments. This setting eliminates the requirements for CAD models and the separated pose estimator training for each object, and thus is more widely applicable for AR applications. OnePose adopts the feature-matching-based visual localization pipeline for this problem setting. It reconstructs sparse object point clouds with SfM [44] and establishes 2D-3D correspondences between keypoints in the query image and the point cloud model to estimate the object pose. Being dependent on detecting repeatable

---

*The first two authors contributed equally. The authors from Zhejiang University are affiliated with the State Key Lab of CAD&CG and the ZJU-SenseTime Joint Lab of 3D Vision.

†Corresponding author.

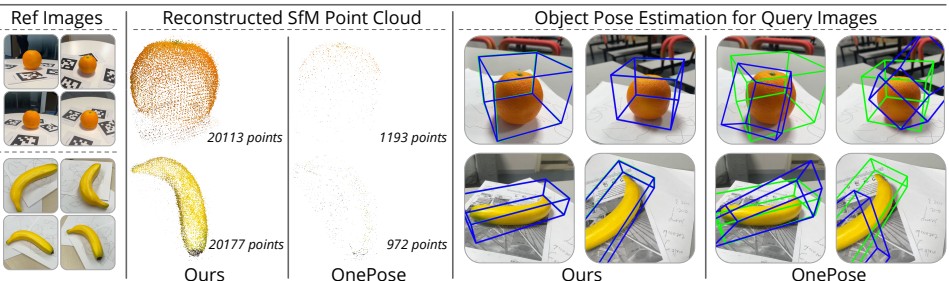

Figure 1: **Comparsion Between Our Method and OnePose [48].** For low-textured objects that are challenging for OnePose, our method can reconstruct their semi-dense point clouds with more complete geometry and thus achieves more accurate object pose estimation. Green and blue boxes represent ground truth and estimated poses, respectively.

keypoints, OnePose struggles with low-textured objects whose complete point clouds are difficult to reconstruct with keypoint-based SfM. Without complete point clouds, pose estimation is prone to failure for many low-textured household objects.

We propose to use a keypoint-free feature matching pipeline on top of OnePose to handle low-textured objects. The keypoint-free semi-dense feature matching method LoFTR [47] achieves outstanding performance on matching image pairs and shows strong capabilities for finding correspondences in low-textured regions. It uses centers of regular grids on a left image as "keypoints", and extracts sub-pixel accuracy matches on the right image in a coarse-to-fine manner. However, this two-view-dependent nature leads to inconsistent "keypoints" and fragmentary feature tracks, which go against the preference of modern SfM systems. Therefore, keypoint-free feature matching cannot be directly applied to OnePose for object pose estimation. We will further elaborate this issue in Sec. 3.1.

To get the best of both worlds, we devise a novel system to adapt keypoint-free matching for one-shot object pose estimation. We propose a two-stage pipeline for reconstructing a 3D structure, striving for both accuracy and completeness. For testing, we propose a sparse-to-dense 2D-3D matching network that efficiently establishes accurate 2D-3D correspondences for pose estimation, taking full advantage of our keypoint-free design.

More specifically, to better adapt LoFTR [47] for SfM, we design a coarse-to-fine scheme for accurate and complete semi-dense object reconstruction. We disassemble the coarse-to-fine structure of LoFTR and integrate them into our reconstruction pipeline. In the coarse reconstruction phase, we use less accurate yet repeatable LoFTR coarse correspondences to construct consistent feature tracks for SfM and yield an inaccurate but complete semi-dense point cloud. Then, our novel refinement phase optimizes the initial point cloud by refining "keypoint" locations in coarse feature tracks to sub-pixel accuracy. As shown in Fig. 1, our framework can reconstruct accurate and complete semi-dense point clouds even for low-textured objects, which lays the foundation for building high-quality 2D-3D correspondences for pose estimation.

At test time, we draw inspiration from the sparse-to-dense matching strategy in visual localization [12], and further adapt it to direct 2D-3D matching in a coarse-to-fine manner for efficiency. Additionally, we use self- and cross-attention to model long-range dependencies required for robust 2D-3D matching and pose estimation of complex real-world objects, which usually contain repetitive patterns or low-textured regions.

We evaluate our framework on the OnePose [48] dataset and the LINEMOD [16] dataset. The experiments show that our method outperforms all existing one-shot pose estimation methods [48, 33] by a large margin and even achieves comparable results with instance-level methods [39, 29] which are trained for each object instance with a CAD model. To further evaluate and demonstrate the capability of our method in real-world scenarios, we collect a new dataset named OnePose-LowTexture, which comprises 80 sequences of 40 low-textured objects.

## Contributions.

- A keypoint-free SfM method for semi-dense reconstruction of low-textured objects.
- A sparse-to-dense 2D-3D matching network for accurate object pose estimation.
- A challenging real-world dataset OnePose-LowTexture composed of 40 low-textured objects with ground-truth object pose annotations.

## 2 Related work

**CAD-Model-Based Object Pose Estimation.** Many previous methods leverage known object CAD models for pose estimation, which can be further categorized into instance-level, category-level, and generalizable methods by their generalizability. Instance-level methods estimate object poses either by directly regressing poses from images [58, 20, 29] or construct 2D-3D correspondences and then solve poses with PnP [39, 59]. The primary deficiency is that these methods need to train a separate network for each object. Category-level methods, such as [55, 51, 21, 54, 56, 4], learn the shape prior shared within a category and eliminate the need for CAD models in the same category at test time. However, these methods cannot handle objects in unseen categories. Some recent methods leverage the generalization power of 2D feature matching for the pose estimation of unseen objects. Reference images are rendered with CAD models and then matched with the query image, using either sparse keypoints matching [61] or dense optical flow [45]. All methods mentioned above depend on high-fidelity textured CAD models for training or rendering, which are not easily accessible in real-world applications. Our framework, instead, reconstructs a 3D object model from pose-annotated images for object pose estimation.

**CAD-Model-Free Object Pose Estimation.** Some recent methods get rid of the CAD model completely. RLLG [2] uses correspondences between image pairs as supervision for training, without a known object model. However, it still requires accurate object masks as supervision, which are not easily accessible without a CAD model. NeRF-Pose [24] reconstructs a neural representation NeRF [36] for an object first and train an object coordinate regression network for pose estimation. These methods are not generalizable to unseen objects. The recently proposed Gen6D [33] and OnePose [48] only require a set of reference images with annotated poses to estimate object poses and can generalize to unseen objects. Gen6D uses detection and retrieval to initialize the pose of a query image and then refine it by regressing the pose residual. However, Gen6D requires an accurately detected 2D bounding box for pose initialization and struggles with occlusion scenarios. OnePose reconstructs objects' sparse point cloud and then extracts 2D-3D correspondences for solving poses. It performs poorly on low-textured objects because of its reliance on repeatably detected keypoints. Our work is inspired by OnePose but eliminates the need for keypoints in the entire framework, which leads to better performance on both textured and low-textured objects.

Notably, leveraging feature matching for object pose estimation is a long-studied problem. Some previous methods [9, 35, 14, 17, 46] extract keypoints on the query image first and perform matching with reference images or SfM model to obtain 2D-3D matches for pose estimation. The main challenges are the ambiguous matches incurred by low-textured and repetitive patterns. They either rely on the ratio test in the matching stage [35, 14, 46] or leverage prioritized hypothesis testing in the outlier filtering stage [9, 17] to reject ambiguous matches. Different from them, our framework eliminates the keypoint detection for the query image by directly performing matching between the 2D feature map and the 3D model, which benefits pose estimation for low-textured objects. Moreover, we leverage the attention mechanism to disambiguate 2D and 3D features for matching, while the direct feature disambiguation is not explored by these methods.

**Structure from Motion and Visual Localization.** Visual localization estimates camera poses of query images relative to a known scene structure. The scene structure is usually reconstructed with Structure-from-Motion (SfM) [44], relying on the feature matching methods [34, 6, 62, 25, 41, 13]. The localization problem is then solved by finding 2D-3D correspondences between the 3D scene model and a query image. HLoc [40] scales up visual localization in a coarse-to-fine manner. It is based on image retrieval and establishes 2D-3D correspondences by lifting 2D-2D matches between query images and retrieved database images to 3D space. However, HLoc is not suitable for our setting since it is slow during pose estimation because it depends on 2D-2D matching of multiple image pairs as the proxy to locate one query image. Our framework is more relevant to the previous visual localization methods which are based on efficient direct 2D-3D matching [43, 26, 49, 32, 3, 60, 27, 5, 7, 50, 42, 12]. To boost matching efficiency between the large-scale point cloud and query images, some of them narrow the searching range by leveraging the priors [43, 26, 5] or compress 3D models by quantizing features [43, 32, 42]. However, these strategies contribute little to the disambiguation of features. They often use priors [43, 32, 27, 42] or geometric verification [49, 3, 60, 50] in the outlier filtering stage to cope with the challenges from low-textured regions or repetitive patterns. In contrast, our method works on the 2D-3D matching phase but focuses on disambiguating features

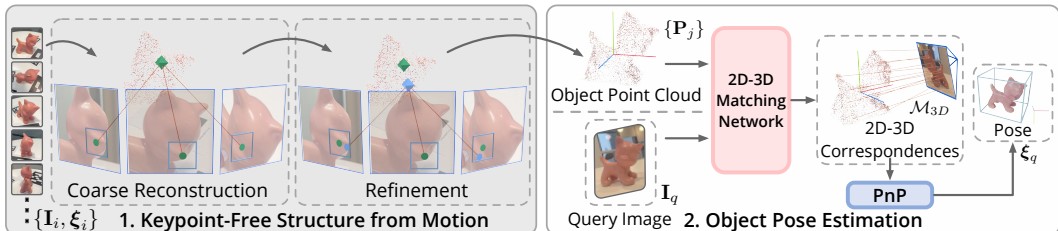

Figure 2: **Overview. 1.** For each object, given a reference image sequence $\{\mathbf{I}_i\}$ with known object poses $\{\boldsymbol{\xi}_i\}$, our keypoint-free SfM framework reconstructs the semi-dense object point cloud in a coarse-to-fine manner. The coarse reconstruction yields the initial point cloud (♦) which is then optimized to obtain an accurate point cloud (♦) in the refinement phase. **2.** At test time, our 2D-3D matching network directly matches a reconstructed object point cloud with a query image $\mathbf{I}_q$ to build 2D-3D correspondences $\mathcal{M}_{3D}$, and then the object pose $\boldsymbol{\xi}_q$ is estimated by solving PnP with $\mathcal{M}_{3D}$.

before matching. Our idea is to directly disambiguate and augment dense features by implicitly encoding their spatial information and relations with other features in a learning-based manner.

Our keypoint-free SfM framework is related to SfM refinement methods PatchFlow [8] and PixSfM [31]. They improve keypoint-based SfM for more accurate 3D reconstructions by refining inaccurately-detected sparse local features with local patch flow [8] or dense feature maps [31]. Different from them, we leverage fine-level matching with Transformer [53] to refine the 2D locations of coarse feature tracks and then optimize the 3D model with geometric error. Please refer to the supplementary for more detailed discussions. [57] also works on keypoint-free SfM. However, it refines the coarse matches by the keypoint relocalization, which is single-view dependent and faces the issue of inaccurate keypoint detection. In contrast, our refinement leverage two-view patches to find accurate matches. Note that there are also methods proposed by keypoint-free matchers [47, 62] to adapt themselves for SfM. They either round matches to grid level [47] or merge matches within a grid to the average location [62] to obtain repeatable "keypoints" for SfM. However, all these approaches trade-off point accuracy for repeatability. On the contrary, our framework obtains repeatable features while preserving the sub-pixel matching accuracy by the refinement phase.

## 3 Methods

An overview of our method is shown in Fig. 2. Given a reference image sequence with known object poses $\{\mathbf{I}_i, \boldsymbol{\xi}_i\}$, our objective is to estimate the object poses $\{\boldsymbol{\xi}_q\}$ for the test images, where $i$ and $q$ denote the indices of the reference images and test images, respectively. To achieve this goal, we propose a novel two-stage pipeline, which first reconstructs the accurate semi-dense object point cloud from reference images (Section 3.2), and then solves the object pose by building 2D-3D correspondences in a coarse-to-fine manner for test images (Section 3.3). Since our method is highly related to the keypoint-free matching method LoFTR [47], we give it a short overview in Section 3.1.

### 3.1 Background

**Keypoint-Free Feature Matching Method LoFTR [47].**  Without a keypoint detector, LoFTR builds semi-dense matches between image pairs (noted as left and right images) in a coarse-to-fine pipeline. First, dense matches between two coarse-level feature maps (⅛ resolution in LoFTR) are built and upsampled, yielding coarse semi-dense matches in the original resolution. With the locations of all left matches fixed, the right matches are refined to a sub-pixel level using fine-level feature maps. Thanks to the keypoint-free design and the global receptive field of Transformers, LoFTR is capable of building correspondences in low-textured regions.

**Problem of Using LoFTR for Keypoint-Based SfM.**  Directly combining LoFTR with modern keypoint-based SfM systems such as COLMAP [44] is not applicable since they rely on fixed keypoints detected on each image to construct feature tracks for estimating 3D structures. However, for LoFTR, its matching locations on a right image depend on its pairing left images. Therefore, the right matching locations are not consistent when paired with multiple left images. Due to this reason, keypoint-free feature matching cannot establish feature tracks across multiple views for effective 3D structure optimization in SfM and is thus not directly applicable in OnePose.

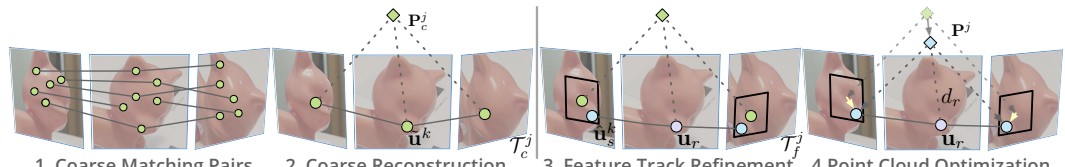

| 1. Coarse Matching Pairs | 2. Coarse Reconstruction | 3. Feature Track Refinement | 4.Point Cloud Optimization |

Figure 3: **Keypoint-Free SfM. 1.** We first build repeatable coarse semi-dense 2D matches between image pairs. **2.** Then, we feed coarse matches to COLMAP [44] to build a coarse feature track $\mathcal{T}_c^j$ and a coarse 3D point $\mathbf{P}_c^j$ ($\diamond$). **3.** To refine $\mathcal{T}_c^j$, we fix a reference node $\mathbf{u}_r$ ($\bigcirc$) and search around the local window ($\square$) of each source node $\tilde{\mathbf{u}}_s^k$ ($\bigcirc$) for sub-pixel correspondences $\hat{\mathbf{u}}_s^k$ ($\bigcirc$). **4.** Finally, we optimize the depth $d_r$ of $\mathbf{u}_r$ by minimizing reprojection errors. We back-project $\mathbf{u}_r$ with its refined $d_r$ to the object coordinate to obtain an optimized accurate object point cloud $\mathbf{P}^j$ ($\diamond$).

## 3.2 Keypoint-Free Structure from Motion

To better adapt LoFTR for SfM, we design a coarse-to-fine SfM framework leveraging the properties of LoFTR's coarse and fine stages separately. Our framework constructs the coarse structure of the feature tracks $\{\mathcal{T}_c^j\}$ and point cloud $\{\mathbf{P}_c^j\}$ in the coarse reconstruction phase. Then in the refinement phase, the coarse structures are refined to obtain the accurate point cloud $\{\mathbf{P}^j\}$. For clarity, in this part, we use $\tilde{\cdot}$ to denote the coarse matching results and use $\hat{\cdot}$ to denote fine matching results. We consider the feature track $\mathcal{T}^j = \{\mathbf{u}^k \in \mathbb{R}^2 | k = 1...N_j\}$ as a set of matched 2D points observing a 3D point $\mathbf{P}^j \in \mathbb{R}^3$. $j$ denotes the index of the feature track and its corresponding 3D point.

**Coarse Reconstruction.** We first strive for the completeness of the initially reconstructed 3D structure. We propose to use the inaccurate yet repeatable coarse correspondences of LoFTR for COLMAP [44] to reconstruct the coarse 3D structure. The coarse correspondences, as shown in Fig. 3 (**1**), can be seen as pixel-wise dense correspondences on downsampled image pairs. Every pixel in the downsampled image can be regarded as a "keypoint" in the original image. Therefore, performing coarse matching can provide repeatable semi-dense correspondences for COLMAP to reconstruct coarse feature tracks $\{\mathcal{T}_c^j\}$ and semi-dense point cloud $\{\mathbf{P}_c^j\}$, as shown in Fig. 3 (**2**).

**Refinement.** Due to the limited accuracy of performing matching on downsampled images, the point cloud from the coarse reconstruction is inaccurate and thus insufficient for the object pose estimation. Therefore, we further refine the object point cloud $\{\mathbf{P}_c^j\}$ with sub-pixel correspondences. To achieve this, we first fix the position of one node for each feature track $\mathcal{T}_c^j$ and refine other nodes within the track. Then, we use the refined tracks $\{\mathcal{T}_f^j\}$ to optimize the $\{\mathbf{P}_c^j\}$.

For the refinement of $\{\mathcal{T}_c^j\}$, we draw the idea from the fine-level matching module in LoFTR and adapt it to the multi-view scenario. As shown in Fig. 3 (**3**), we first select and fix one node in each $\mathcal{T}_c^j$ as the reference node $\mathbf{u}_r$, and then perform fine matching with each of the remaining source node $\tilde{\mathbf{u}}_s^k$. The fine matching searches within a local region around each $\tilde{\mathbf{u}}_s^k$ for a sub-pixel correspondence $\hat{\mathbf{u}}_s^k$, so the nodes' locations in the coarse feature track are refined. We denote the refined feature tracks as $\{\mathcal{T}_f^j\}$. Details about the selection of reference nodes $\mathbf{u}_r$ are provided in the supplementary material.

We now treat the refined feature tracks $\{\mathcal{T}_f^j\}$ as fixed measurements, and optimize the 3D locations of the coarse point cloud $\{\mathbf{P}_c^j\}$ using reprojection errors as shown in Fig. 3 (**4**). To accelerate the convergence, inspired by SVO [11], we further decrease the DoF of each $\mathbf{P}_c^j$ by only optimizing the depth $d_r$ of each reference node $\mathbf{u}_r$. Specifically, we transform each point $\mathbf{P}_c^j$ to the frame of $\mathbf{u}_r$ and use its coordinate of $z$-axis to initialize $d_r$. Then we optimize each reference node depth $d_r$ by minimizing the distance between each reprojected location and the refined feature location $\hat{\mathbf{u}}_s^k$:

$$d_r^* = \underset{d_r}{\arg\min} \sum_{k \in N_j - 1} \|\hat{\mathbf{u}}_s^k - \boldsymbol{\pi} \left(\boldsymbol{\xi}_{r \to s_k} \cdot \boldsymbol{\pi}^{-1}(\mathbf{u}_r, d_r)\right)\|^2. \tag{1}$$

where $\boldsymbol{\pi}$ is the projection determined by intrinsic camera parameters, and $\boldsymbol{\xi}_{r \to s_k} = \boldsymbol{\xi}_{s_k} \cdot \boldsymbol{\xi}_r^{-1}$ is the relative pose between the frame of the reference node and $k$-th source node.

Finally, the optimized depth $d_r^*$ of each reference node is transformed to the canonical object coordinate to get the refined 3D point $\mathbf{P}^j$. Notably, when applying the proposed system in practical AR applications, we can optimize inaccurate camera poses obtained from ARKit along with the 3D points, i.e., solving a bundle adjustment problem. For the later 2D-3D matching at test time, we calculate and store each 3D point feature by averaging the 2D features of its associated 2D points.

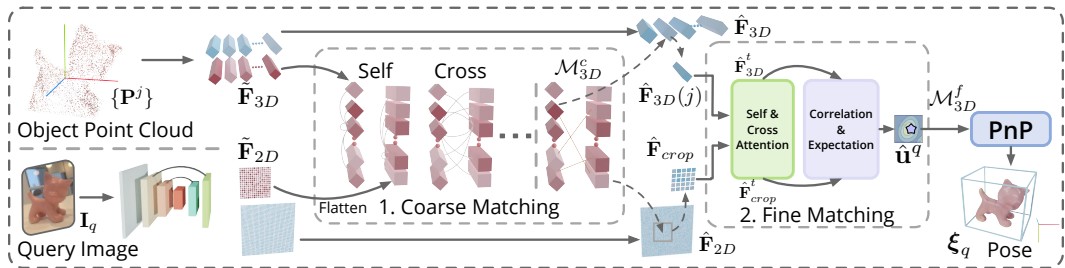

Figure 4: **Object Pose Estimation.** At test time, we first extract multi-scale query image features $\{\tilde{\mathbf{F}}_{2D}, \hat{\mathbf{F}}_{2D}\}$. **Coarse Matching** module transforms coarse 2D and 3D features $N_c$ times with self- and cross-attention modules and then build their coarse 2D-3D correspondences $\mathcal{M}_{3D}^c$. Next, we crop the local window $\hat{\mathbf{F}}_{crop}$ on the fine feature map around each coarse 2D match. **Fine Matching** module transforms the 3D feature and cropped 2D features and calculates each 2D fine match location $\hat{\mathbf{u}}^q$ with feature correlation and expectation. The object pose $\boldsymbol{\xi}_q$ is then solved using PnP with $\mathcal{M}_{3D}^f$.

Note that we store coarse and fine 3D features separately, which are extracted from multi-resolution feature maps of LoFTR's feature backbone.

### 3.3 Object Pose Estimation

At test time, we establish 2D-3D matches between the object point cloud $\{\mathbf{P}^j\}$ and the query image $\mathbf{I}_q$ to estimate object pose $\boldsymbol{\xi}_q$. Inspired by [47], we first extract hierarchical feature maps of $\mathbf{I}_q$ and then perform matching in a coarse-to-fine manner for efficiency, as illustrated in Fig. 4.

**Coarse 2D-3D Matching.** We first perform dense matching between the pre-calculated coarse 3D point features $\tilde{\mathbf{F}}_{3D} \in \mathbb{R}^{N \times C_c}$ and the extracted coarse image feature map $\tilde{\mathbf{F}}_{2D} \in \mathbb{R}^{\frac{H}{8} \times \frac{W}{8} \times C_c}$. This phase globally searches for a rough correspondence of each 3D object point in the query image, which also determines whether the 3D point is observable by $\mathbf{I}_q$.

We augment 3D and 2D features $\{\tilde{\mathbf{F}}_{3D}, \tilde{\mathbf{F}}_{2D}\}$ with positional encodings, to make them position-dependent, and thus facilitates their matching. Please refer to the supplementary material for more details. Then we flatten the 2D feature map and apply self- and cross-attention layers by $N_c$ times to yield the transformed features $\{\tilde{\mathbf{F}}_{3D}^t, \tilde{\mathbf{F}}_{2D}^t\}$. Linear Attention [19] is used in our model to reduce the computational complexity, following [47]. A score matrix S is calculated by the similarity between two sets of features $\tilde{\mathbf{F}}_{3D}^t$ and $\tilde{\mathbf{F}}_{2D}^t$. We then apply dual-softmax operation [52] on S to get the matching probability matrix $\mathcal{P}^c$:

$$\mathcal{P}^c(j,q) = \text{softmax}\left(\text{S}\left(j,\cdot\right)\right)_q \cdot \text{softmax}\left(\text{S}\left(\cdot,q\right)\right)_j, \text{ where } \text{S}\left(j,q\right) = \frac{1}{\tau} \cdot \langle \tilde{\mathbf{F}}_{3D}^t(j), \tilde{\mathbf{F}}_{2D}^t(q)\rangle. \tag{2}$$

$\langle\cdot,\cdot\rangle$ is the inner product, $\tau$ is a scale factor, and $j$ and $q$ denote the indices of a 3D point and a pixel in the flattened query image, respectively. The coarse 2D-3D correspondences $\mathcal{M}_{3D}^c$ are established from $\mathcal{P}^c$ by selecting correspondences above a confidence threshold $\theta$:

$$\mathcal{M}_{3D}^c = \{(j,q) \mid \forall (j,q) \in \text{MNN}\left(\mathcal{P}^c\right), \mathcal{P}^c\left(j,q\right) \geq \theta\}. \tag{3}$$

MNN refers to finding the mutual nearest neighbors. We use this strict criterion to suppress potential false matches.

**Fine Matching.** For a visible 3D point $\mathbf{P}^j$ determined by $\mathcal{M}_{3D}^c$, our fine matching module searches for its sub-pixel 2D correspondence $\hat{\mathbf{u}}^q$ within the local region of its coarse correspondence $\tilde{\mathbf{u}}^q$. Similar to [47], we crop a local window $W$ with a size of $w \times w$ around $\tilde{\mathbf{u}}^q$ in the fine feature map $\hat{\mathbf{F}}_{2D} \in \mathbb{R}^{\frac{H}{2} \times \frac{W}{2} \times C_f}$. Then the cropped feature map $\hat{\mathbf{F}}_{crop} \in \mathbb{R}^{w \times w \times C_f}$ and corresponding 3D fine feature $\hat{\mathbf{F}}_{3D}(j) \in \mathbb{R}^{C_f}$ are transformed by $N_f$ self- and cross- attention layers. We correlate the transformed 3D fine feature vector $\hat{\mathbf{F}}_{3D}^t$ with all elements in the transformed 2D feature $\hat{\mathbf{F}}_{crop}^t$ and apply a softmax to get the probability distribution of its 2D correspondence in the cropped local window:

$$p(\mathbf{u}|j, \hat{\mathbf{F}}_{3D}^t, \hat{\mathbf{F}}_{crop}^t) = \frac{\exp{(\hat{\mathbf{F}}_{3D}^t(j)^{\text{T}} \cdot \hat{\mathbf{F}}_{crop}^t(\mathbf{u}))}}{\sum_{\mathbf{u} \in W} \exp{(\hat{\mathbf{F}}_{3D}^t(j)^{\text{T}} \cdot \hat{\mathbf{F}}_{crop}^t(\mathbf{u}))}}. \tag{4}$$

The fine correspondence $\hat{\mathbf{u}}^q$ of $\mathbf{P}^j$ on the query image is then obtained with an expectation:

$$\hat{\mathbf{u}}^q = \tilde{\mathbf{u}}^q + \sum_{\mathbf{u} \in W} \mathbf{u} \cdot p(\mathbf{u}|j, \hat{\mathbf{F}}_{3D}^t, \hat{\mathbf{F}}_{crop}^t). \tag{5}$$

After building the 2D-3D correspondences $\mathcal{M}_{3D}^f$ between the query image and the object point cloud, we solve the object pose $\boldsymbol{\xi}_q$ with the Perspective-n-Point (PnP) [22] algorithm and RANSAC [10].

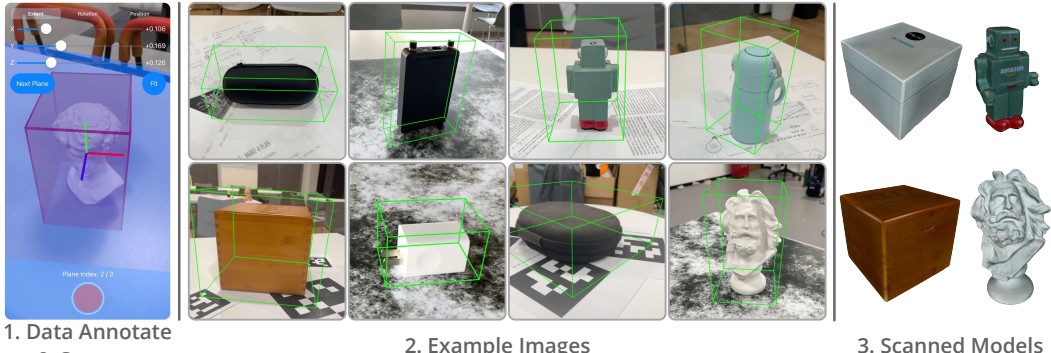

| 1. Data Annotate & Capture | 2. Example Images | 3. Scanned Models |

Figure 5: Data capture and example images of the proposed OnePose-LowTexture dataset.

**Supervision.** We jointly train the coarse and fine modules of our 2D-3D matching framework with different supervisions, following [47]. The ground-truth 2D-3D correspondences $\mathcal{M}_{3D}^{gt}$ and the coarse matching probability matrix $\mathcal{P}_{gt}^c$ are obtained by projecting observable SfM points to the 2D frame with the ground-truth object pose. We optimize the coarse matching module by minimizing the focal loss [30] between the predicted $\mathcal{P}^c$ and $\mathcal{P}_{gt}^c$. For the fine module, we minimize the $\ell_2$ loss between the predicted 2D coordinate $\hat{\mathbf{u}}^q$ and the ground truth $\hat{\mathbf{u}}_{gt}^q$. The total loss is the weighted sum of coarse and fine losses. More details are provided in the supplementary material.

### 3.4 Implementation Details

**Keypoint-Free SfM.** The COLMAP [44] triangulation is used to construct the coarse 3D structure. In the refinement phase, the local window with a size of $9 \times 9$ is searched for the sub-pixel correspondence of each reference node. The Levenberg-Marquardt algorithm [23] implemented in DeepLM [18] is used to optimize the coarse object point cloud. We use the LoFTR [47] outdoor model pre-trained on MegaDepth [28]. Running time analyses are given in the supplementary material.

**Object Pose Estimation.** We use ResNet-18 [15] as the image backbone and set $N_c = 3, N_f = 1$ for the 2D-3D attention module. The scale factor $\tau$ is 0.08, the cropped window size $w$ in the fine level is 5, and the confidence threshold $\theta$ is set to 0.4. As for training, the backbone of our model is initialized with the LoFTR outdoor model, and the remaining parts of our model use randomly initialized weights. The entire model is trained on the OnePose training set, and we randomly sample or pad the reconstructed point cloud to 7000 points for training. We use the AdamW optimizer with an initial learning rate of $4 \times 10^{-3}$. The network training takes about 20 hours with a batch size of 32 on 8 NVIDIA-V100 GPUs. During testing, we use all reconstructed 3D points ($\sim 15000$) for building 2D-3D correspondences, and our 2D-3D matching module takes 88ms for a $512 \times 512$ query image on a single V100 GPU.

## 4 Experiments

### 4.1 Datasets

**OnePose and LINEMOD Datasets.** We validate our method on the OnePose [48] and LINEMOD [16] datasets. The OnePose dataset is newly proposed, which contains around 450 real-world video sequences of 150 objects. LINEMOD is a broadly used dataset for object pose estimation. For both datasets, we follow the train-test split in previous methods [48, 29].

**OnePose-LowTexture Dataset.** Since the original OnePose evaluation set mainly comprises textured objects, we collected an additional test set, named OnePose-LowTexture, to supplement the original OnePose dataset. The proposed dataset is composed of 40 household low-textured objects. For each object, there are two corresponding videos captured with different backgrounds, one as the reference video and the other for testing. Besides, to evaluate and compare our method with CAD-model-based methods, we further obtain high-fidelity 3D models of eight randomly selected objects with a commercial 3D scanner. Some example images are shown in Fig. 5. Please refer to the supplementary material for more details.

Table 1: **Comparison with *One-shot* Baselines.** Our method is compared with HLoc [40] combined with different feature matching methods and OnePose [48], using the *cm-degree* pose success rate with different thresholds.

| | OnePose dataset | | | OnePose-LowTexture | | | Time (ms) |
|---|---|---|---|---|---|---|---|
| | 1cm-1deg | 3cm-3deg | 5cm-5deg | 1cm-1deg | 3cm-3deg | 5cm-5deg | |
| HLoc *(SPP + SPG)* | **51.1** | 75.9 | 82.0 | 13.8 | 36.1 | 42.2 | 835 |
| HLoc *(LoFTR*)* | 39.2 | 72.3 | 80.4 | 13.2 | 41.3 | 52.3 | 909 |
| OnePose | 49.7 | 77.5 | 84.1 | 12.4 | 35.7 | 45.4 | **66.4** |
| Ours | **51.1** | **80.8** | **87.7** | **16.8** | **57.7** | **72.1** | 88.2 |

Table 2: **Comparison with *Instance-level* Baseline.** Our method is compared with PVNet[39] on objects with CAD models in the OnePose-LowTexture dataset using the *ADD(S)-0.1d* metric.

| Obj. ID | 0700 | 0706 | 0714 | 0721 | 0727 | 0732 | 0736 | 0740 | Avg. |
|---|---|---|---|---|---|---|---|---|---|
| PVNet | 12.3 | 90.0 | 68.1 | 67.6 | 95.6 | 57.3 | 49.6 | **61.3** | 62.7 |
| Ours | **89.5** | **99.1** | **97.2** | **92.6** | **98.5** | **79.5** | **97.2** | 57.6 | **88.9** |

## 4.2 Experiment Settings and Baselines

**Baselines.** We compare the proposed method with the following baselines in two categories: 1) *One-shot baselines* [48, 33, 40] that hold the same setting as ours. OnePose [48] and HLoc [40] are most relevant to our method in leveraging feature matching for reconstruction and pose estimation. To be specific, we compare with HLoc combined with different feature matching methods including SuperGlue [41] and LoFTR [47]. 2) *Instance-level baselines* [39, 29] that require CAD-models and need to be trained separately for each object. These methods achieve high accuracy through training on many rendered images with extensive data augmentation. We compare our method with them to demonstrate that our method achieves competitive results while not relying on CAD models and eliminating per-object pose estimator training.

**Evaluation Protocols.** We compare our method with OnePose and HLoc using the same set of reference images. Since HLoc's original retrieval module is designed for the outdoor scenes, we use uniformly sampled 10 reference views for 2D-2D matching for pose estimation, following [48]. For the comparison with PVNet [39], we follow its original training setting, which first samples 8 keypoints on the object surface and then trains a network using 5000 synthetic images for each object. In contrast, our method only uses around 200 reference images to reconstruct the object point cloud. We evaluate our method and PVNet on the same real-world test sequences, while our matching model has never seen the test objects before. As for the experiments on LINEMOD, we compare our method with OnePose by running their open-source code. Our method and OnePose share the same 2D bounding boxes from an off-the-shelf object detector YOLOv5 [1]. Note that the object detector is trained on real-world images only to provide rough bounding boxes. We use the real training images ($\sim 180$) for object reconstruction and all test images for evaluation. The results of other baselines on LINEMOD are from the original papers.

**Metrics.** We use metrics including the *cm-degree* pose success rate, the *ADD(S)-0.1d* average distance with a threshold of $10\%$ of the object diameter, and the 2D projection error *Proj2D* with a threshold of 5 pixels. The definitions of these metrics are detailed in the supplementary material.

## 4.3 Results on the OnePose and OnePose-LowTexture Datasets

**Comparison with *One-shot* Baselines.** The *cm-degree* success rate with different thresholds are used for evaluation. As shown in Tab. 1, our method substantially outperforms OnePose [48] and HLoc [40]. Objects in the OnePose dataset have rich textures, benefiting keypoint detection. Therefore, keypoint-based methods OnePose and HLoc (*SPP+SPG*) perform reasonably well. Our method achieves even higher accuracy thanks to the keypoint-free design, effectively utilizing both texture-rich and low-textured object regions for pose estimation. On the OnePose-LowTexture dataset, our method surpasses OnePose and HLoc by a large margin. This further demonstrates the capability of our keypoint-free design for object reconstruction and the sparse-to-dense 2D-3D matching for object pose estimation. HLoc (*LoFTR*) uses LoFTR coarse matches for SfM and uses full LoFTR to match the query image and its retrieved images for pose estimation. It does not rely on keypoints, similar to our design. Our method significantly outperforms it on accuracy and runs $\sim 10\times$ faster.

Table 3: **Results on LINEMOD.** Our method is compared with *Instance-level* and *One-shot* baselines. Note that Gen6D is fine-tuned on a selected subset of objects and uses the rest for testing. Gen6D[†] is the version without fine-tuning on LINEMOD. Symmetric objects are indicated by *.

| Type | Name | ape | benchwise | cam | can | cat | driller | duck | eggbox* | glue* | holepuncher | iron | lamp | phone | Avg. |
|---|---|---|---|---|---|---|---|---|---|---|---|---|---|---|---|
| | | | | | | | *ADD(S)-0.1d* | | | | | | | | |
| Instance-level | CDPN | 67.3 | 98.8 | 92.8 | 96.6 | 86.6 | 95.1 | 75.2 | 99.6 | 99.6 | 89.7 | 97.9 | 97.8 | 80.7 | 91.4 |
| | PVNet | 43.6 | 99.9 | 86.9 | 95.5 | 79.3 | 96.4 | 52.6 | 99.2 | 95.7 | 81.9 | 98.9 | 99.3 | 92.4 | 86.3 |
| One-shot | Gen6D[†] | - | 62.1 | 45.6 | - | 40.9 | 48.8 | 16.2 | - | - | - | - | - | - | - |
| | Gen6D | - | 77.0 | 66.1 | - | 60.7 | 67.4 | 40.5 | 95.7 | 87.2 | - | - | - | - | - |
| | OnePose | 11.8 | 92.6 | 88.1 | 77.2 | 47.9 | 74.5 | 34.2 | 71.3 | 37.5 | 54.9 | 89.2 | 87.6 | 60.6 | 63.6 |
| | Ours | 31.2 | 97.3 | 88.0 | 89.8 | 70.4 | 92.5 | 42.3 | 99.7 | 48.0 | 69.7 | 97.4 | 97.8 | 76.0 | 76.9 |
| | | | | | | | *Proj2D* | | | | | | | | |
| Instance-level | CDPN | 97.5 | 98.8 | 98.6 | 99.6 | 99.3 | 94.9 | 98.4 | 99.1 | 98.4 | 99.5 | 97.9 | 95.7 | 96.8 | 98.0 |
| | PVNet | 99.2 | 99.8 | 99.2 | 99.9 | 99.3 | 96.9 | 98.0 | 99.3 | 98.5 | 100.0 | 99.2 | 98.3 | 99.4 | 99.0 |
| One-shot | OnePose | 35.2 | 94.4 | 96.8 | 87.4 | 77.2 | 76.0 | 73.0 | 89.9 | 55.1 | 79.1 | 92.4 | 88.9 | 69.4 | 78.1 |
| | Ours | 97.3 | 99.6 | 99.6 | 99.2 | 98.7 | 93.1 | 97.7 | 98.7 | 51.8 | 98.6 | 98.9 | 98.8 | 94.5 | 94.3 |

The improved accuracy and speed come from the accurate point cloud reconstructed by our novel SfM framework and the efficient 2D-3D matching module.

**Comparison with *Instance-level* Baseline PVNet.**  On the OnePose-LowTexture dataset, the proposed method is compared with PVNet [39] on the subset objects with scanned models. The *ADD(S)-0.1d* results are presented in Tab. 2. Even though PVNet is trained on a large number ($\sim 5000$) of rendered images covering almost all possible views, our method still outperforms it on most objects without additional training. We attribute this to PVNet's susceptibility to domain gaps and our matching module's robustness and generalizability, thanks to its large-scale pre-training.

### 4.4  Results on LINEMOD

We compare the proposed method with OnePose [48] and Gen6D [33] which are under the *One-shot* setting, and *Instance-level* methods PVNet [39] and CDPN [29] on *ADD(S)-0.1d* and *Proj2D* metrics. As shown in Tab. 3, our method outperforms existing one-shot baselines significantly and achieves comparable performance with instance-level methods. Notably, our method and OnePose are only trained on the OnePose training set and tested on LINEMOD without additional training.

Since LINEMOD is mainly composed of low-textured objects, our method outperforms OnePose significantly thanks to the keypoint-free design. Gen6D [33] is CAD-model-free and can generalize to unseen objects similar to our method. However, it relies on detecting accurate object bounding boxes for pose initialization, which is hard on LINEMOD because of the poor image quality and slight object occlusion. In contrast, our method only needs rough object detection to reduce possible mismatches, which is more robust to detection error. Moreover, the performance of Gen6D drops significantly without training on a subset of LINEMOD, while our method requires no extra training and achieves much higher accuracy than Gen6D. The experiment demonstrates the superiority of our method over existing methods under the one-shot setting.

Our method has lower or comparable performance with instance-level methods  [39, 29], which are trained to fit each object instance, and thus perform well naturally, at the expense of the tedious training for each object. In contrast, our method is grounded in highly generalizable local features and generalizes to unseen objects with comparable performances.

### 4.5  Ablation Studies

We conduct several experiments on the OnePose dataset and the OnePose-LowTexture dataset to validate the efficacy of the point cloud refinement in the SfM framework and the attention module in our 2D-3D matching module. More ablation studies are detailed in the supplementary material.

**Point Cloud Refinement in the Keypoint-Free SfM.**  We validate the effectiveness of the point cloud refinement from two perspectives, as shown in Tab. 4 (Left). One is evaluating the accuracy of the reconstructed point cloud with ground truth object mesh on OnePose-LowTexture dataset following evaluations in [31], and the other is quantifying the impact of reconstruction accuracy on the *cm-degree* pose success rate. Compared with the coarse point cloud reconstructed with coarse-level LoFTR only, the accuracy of our refined point cloud increased significantly, especially under a strict threshold (*1mm*). Moreover, using the refined point cloud for object pose estimation brings around

Table 4: **Ablation Studies.** We quantitatively validate the effectiveness of the point cloud refinement in the keypoint-free SfM and the attention module in the 2D-3D matching network, using the point cloud accuracy metric and *cm-degree* pose success rate with different thresholds.

| | Point Cloud Accuracy | | | Pose Success Rate on OnePose Dataset | | | | Pose Success Rate on OnePose-LowTexture | | |
| | 1mm | 3mm | 5mm | 1cm-1deg | 3cm-3deg | 5cm-5deg | | 1cm-1deg | 3cm-3deg | 5cm-5deg |
|---|---|---|---|---|---|---|---|---|---|---|
| w/o refine. | 26.6 | 71.2 | 85.7 | 43.9 | 78.3 | 85.9 | w/o attention. | 12.2 | 40.7 | 55.3 |
| w refine. | **30.9** | **75.8** | **87.7** | **51.1** | **80.8** | **87.7** | w attention. | **16.8** | **57.7** | **72.1** |

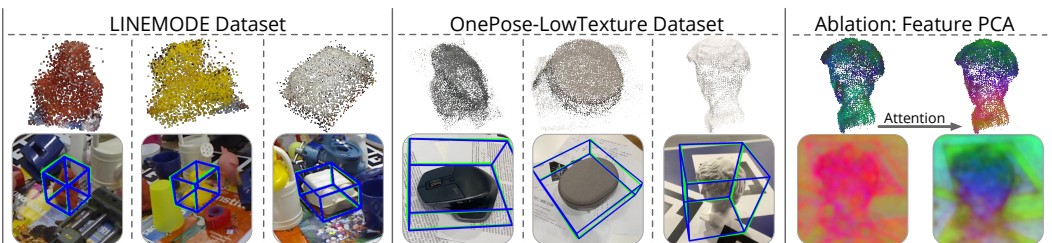

Figure 6: **Qualitative Results** showing the reconstructed semi-dense object point clouds and the estimated object poses. The ablation part visualizes the 2D and 3D features before and after our 2D-3D attention module. Features become more discriminative as shown by the color contrast.

7% improvement on the strict *1cm-1deg* metric. These experiments demonstrate that the point cloud refinement improves the reconstructed point clouds' precision, thus benefiting the pose estimation.

**Attention module in the Pose Estimation Network.**    We validate the attention design in our matching network quantitatively and qualitatively. It is shown in Tab. 4 (Right) that compared with directly matching the backbone features, using the transformed features for 2D-3D correspondences obtains 15% improvement on the *5cm-5deg* metric on OnePose-LowTexture dataset. The visualization of features in Fig. 6 shows that the transformed 2D and 3D features become more discriminative for establishing correspondences. The ablation study demonstrates that the attention module provides the global receptive field and plays a critical role in the pose estimation of low-textured objects.

## 5    Conclusion

We propose a keypoint-free SfM and pose estimation pipeline that enables pose estimation of both texture-rich and low-textured objects under the one-shot CAD-model-free setting. Our method can efficiently reconstruct accurate and complete 3D structures of low-textured objects and build robust 2D-3D correspondences with the test image for accurate object pose estimation. The experiments show that our method achieves significantly better pose estimation accuracy compared with existing CAD-model-free methods, and even achieves comparable results with CAD-model-based instance-level methods. Although we do not see the immediate negative societal impact of our work, we do note that accurate object pose estimation can be potentially used for malicious purposes.

**Limitations.**    Being dependent on local feature matching, our method inherently suffers from very low-resolution images and extreme scale and viewpoint changes. In the current pipeline, we still need a separate object detector to provide rough regions of interest. In the future, we envision a more tight integration with the object detector, where object detection can also be carried out through local feature matching.

**Acknowledgements.**    The authors would like to acknowledge the support from the National Key Research and Development Program of China (No. 2020AAA0108901), NSFC (No. 62172364), the ZJU-SenseTime Joint Lab of 3D Vision, and the Information Technology Center and State Key Lab of CAD&CG, Zhejiang University.

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
