# OpenReview forum: "OnePose++: Keypoint-Free One-Shot Object Pose Estimation without CAD Models"
_NeurIPS.cc/2022/Conference — NeurIPS 2022 Accept_

### Official Review · Reviewer_BV2E · 2022-07-06

**Rating:** 6
**Confidence:** 3
**Soundness:** 3 good
**Presentation:** 3 good
**Contribution:** 3 good

**Summary:**

The paper proposes an improvement of the one-shot pose estimation system OnePose to better estimate the pose of low-texture objects. In particular, the keypoint-based matching component of OnePose is replaced with the key-point free matching method LoFTR. For the evaluation of the improved functionality, the authors propose the new dataset OnePose-HARD, which contains low-texture objects along with their pose annotations.


**Questions:**


- Please consider including example images of the OnePose-HARD dataset in the paper.
- Please clarify the source of the results of other methods that you compare your method to: Where are these numbers taken from or did you evaluate the methods yourself? In particular, the results of OnePose on the OnePose dataset and on LINEMOD, as described above. Please also describe why some results are underlined in Table 1.
- Please consider including the runtimes of your method and the methods you compare to, in order to support the claim that your method is 10× faster.


Additional remarks (typos, suggestions etc.), no need to address in the rebuttal:
- line 49: "establish" -> "establishes"
- line 70: "OnePose-HARD ," -> "OnePose-HARD,"
- line 104: "keypoints ." -> "keypoints."
- Figure 2, caption: "a reference image sequences" -> "a reference image sequence"
- Figure 2, caption: "point cloud which are" -> "point cloud which is"
- line 131: "build" -> "builds"
- line 295: "number(~ 5000)" -> "number (~ 5000)"
- On the naming of the proposed dataset: The suffix "HARD" is very generic and does not tell what the difficulties actually are. Please consider a more informative suffix, such as "LowTexture".


**Limitations:**

Yes, limitations and impact were discussed where applicable.

**Strengths And Weaknesses:**

Originality:
+ The idea of replacing keypoint-based matching with keypoint-free matching for improved low-texture performance is straightforward and a logical extension of the prior work.

Quality:
+ The paper is well written and structured.
+ The paper precisely adheres to the formatting requirements and the page limit.
+ Related work is described sufficiently and it is made clear how the proposed method is positioned in the existing landscape.
+ The proposed method is evaluated on appropriate datasets and compared to relevant state-of-the-art methods.

Clarity:
+ The language is clear and easy to follow.
+ Methods used in the paper, e.g., LoFTR, are explained briefly, but well understandable.
- The description of the proposed OnePose-HARD dataset is quite abstract. Example images of the new dataset are not included in the paper.
- In the evaluation, it is not always clear where the results of the other methods come from. E.g., in Table 1, the results of the OnePose method on the OnePose dataset is given as an overall single value for the whole dataset, while the original OnePose paper lists the results of the three categories "large", "medium", and "small" separately, but no overall single value. Similarly, in Table 3, the paper gives results for OnePose on the LINEMOD dataset, but the original OnePose paper contains no such evaluation.
- The authors claim that their method is "~10× faster" [line 290] than other methods, but the actual runtimes of the other methods are not listed in the paper.

Significance:
+ The addressed issue with low-texture objects is relevant in practice.
+ The results of the proposed method seem to be significantly better than comparable methods, particularly in low-texture scenarios.

---

> ### Author Response · Authors · 2022-08-02
> **The response for the major concerns**
>
> We thank the reviewers for the insightful suggestions. We address the major concerns below:
>
> >**Q1:** The description of the proposed OnePose-HARD dataset is quite abstract. Example images of the new dataset are not included in the paper. Please consider including example images of the OnePose-HARD dataset in the paper.
>
> **R1:** Thank you very much for the suggestions. In fact, we describe the details of the OnePose-HARD dataset, and the example images are included in the supplementary L62-78. We will provide more detailed information and example images in the revised paper.
>
> >**Q2:** In the evaluation, it is not always clear where the results of the other methods come from. Please clarify the source of the results of other methods that you compare your method to.
>
> **R2:** We thank the reviewer for pointing out this missing information. We clarify as follows and promise to add to the final paper.
>
> For the evaluation of the OnePose dataset, because of the limited space for writing, we report the overall metric by average over the whole OnePose evaluation set. The overall metric results come from OnePose's supplementary material(https://zju3dv.github.io/onepose/files/onepose_supp.pdf, the first row of results in Tab.2). We believe the overall metric won't affect the comparison.
>
> Additionally, we use underlines to denote the second place results while using bold to denote first place results in Tab.1. We will add the illustrations of these symbols in the caption.
>
> The results of PVNet on the OnePose-HARD dataset are obtained by running their open-source code. Details are located at L267-268.
> For the experiments on the LINEMOD dataset, since OnePose contains no such evaluation, we evaluate the OnePose by running their open-source code and using their pre-trained model. As described in L274-275, the results of other baselines, including PVNet, CDPN, Gen6D, are from their original paper.
>
> >**Q3:** The authors claim that their method is "~10× faster" [line 290] than other methods, but the actual runtimes of the other methods are not listed in the paper. Please consider including the runtimes of your method and the methods you compare to, in order to support the claim that your method is 10× faster.
>
> **R3:** We thank the reviewer for pointing out this missing information. We provide the runtimes of generalizable pose estimators as follows and will add them to Tab.1
> |  Ours|  OnePose|  HLoc(LoFTR)| HLoc(SPP+SPG)|
> |:-    |:-       |:-           |:-            |
> | 87ms |   66ms  |    909ms    |     835ms    |
>
> The runtimes are evaluated on the same server described in L236. As described in L288-291, our method runs ~10× faster than HLoc-based methods. Since we use more 3D points to perform matching with query feature maps in a coarse-to-fine manner, our method is a little slower than OnePose.

---

> > ### Comment · Reviewer_iDEM · 2022-08-05
> > **Re: The response for the major concerns**
> >
> > I have a follow-up question on the OnePose++ dataset: How many training and testing images are there? I can't seem to find this information in the supp. material.

---

> > > ### Author Response · Authors · 2022-08-09
> > > **The response to the follow-up question**
> > >
> > > > **Q1:** I have a follow-up question on the OnePose++ dataset: How many training and testing images are there? I can't seem to find this information in the supp. material.
> > >
> > > Thank you very much for your comments. As described in Line 247-249, the OnePose-HARD is an evaluation set to supplement the original OnePose dataset. We use all of the objects in the OnePose-HARD dataset for testing. The total number of images in the reference sequences is 35521, and the total number of images in the query sequences is 32477. We will add the missing information in the final version.

---

> > > > ### Comment · Reviewer_iDEM · 2022-08-09
> > > > **Re: The response to the follow-up question**
> > > >
> > > > Thank you very much. That answers my question.

---

### Official Review · Reviewer_iDEM · 2022-07-10

**Rating:** 3
**Confidence:** 4
**Soundness:** 2 fair
**Presentation:** 2 fair
**Contribution:** 2 fair

**Summary:**

The paper considers the problem of object pose estimation in scenarios where CAD model of the objects are not available. The paper describes OnePose++, a variant of the recently proposed OnePose approach that uses densely extracted descriptors (via LoFTR) rather than the SuperPoint keypoints used by OnePose. LoFTR provides matches between pairs of images, where the 2D positions of matching points vary depending on the image pair. As the resulting 2D positions are not repeatable, the paper uses a SfM approach designed to handle this scenario in order to build the 3D model of the object used for pose estimation. 2D-3D matches between a query image and the SfM model are established by directly matching descriptors against the 3D model in a coarse-to-fine manner. In addition to the OnePose++ method, the paper introduces a harder variant of the OnePose dataset, named OnePose-HARD. Experimental results show that OnePose++ outperforms most baselines by a wide margin (in particular, OnePose++ consistently outperforms OnePose).

**Questions:**

In order to consider raising my score, I would like to see the following points addressed in a rebuttal:
Q1) Please describe in detail how the proposed keypoint-free SfM approach differs from prior work in this area.
Q2) Why were Patch2Pix, Dual RCNet, etc. not considered as baselines?
Q3) Please describe the relation of OnePose++ to prior work on SfM-based object pose estimation and visual localization (see W3 above).

**Limitations:**

The paper adequately discusses limitations and potential negative social impact.

**Strengths And Weaknesses:**

The paper has multiple strengths:
S1) The proposed OnePose++ approach is a natural extension of OnePose that shows how to swap keypoint-based features with keypoint-free features. Since the latter have shown promise in challenging scenes, e.g., for weakly textured objects or under strong illumination conditions, this is interesting.

S2) OnePose++ clearly outperforms OnePose and also most of the baseline methods. The strong results are a strength of the paper.

S3) The paper provides a detailed ablation study that analyzes the impact of the individual components of OnePose++.

S4) The proposed dataset, OnePose-HARD, seems very challenging and thus has the potential to drive research in the field. It will be of interest to the community.

On the negative side, there are also multiple weaknesses:
W1) Both OnePose and OnePose++ are misclassified as being generalizable object pose estimation approaches. It is true that the underlying LoFTR features and the coarse and fine matching stages generalize beyond the data they were trained on. Yet, OnePose++ requires that "a video sequence with annotated poses is available for each object" is available and builds a SfM model for this particular type of object. I don't see how this is not an instance-level method. The SfM model for one object might give reasonable results for another object if both objects have very similar shapes (and textures). But a single model for one particular type of an object, e.g., a particular chair, will not generalize over the full class (e.g., all potential chairs).
Given that OnePose++ is an instance-level method, it should be more closely compared to other instance-level methods. It is not too surprising that it outperforms the generalizable baselines as it is able to train (in the form of building a SfM model) per object.

W2) The paper claims proposing a "keypoint-free SfM framework for accurate and complete semi-dense reconstruction" as one of its main contributions. It is not clear to my how the described framework contributes novelty to the literature:
a) Besides LoFTR, there are other keypoint-free descriptors, e.g., Patch2Pix [Zhou et al., Patch2Pix: Epipolar-Guided Pixel-Level Correspondences, CVPR'21], Sparse NCNet [Rocco et al., Efficient ´neighbourhood consensus networks via submanifold sparse
convolutions, ECCV'20], and Dual RCNet [Li et al., Dual-Resolution Correspondence Networks, NeurIPS'20], that are evaluated in a visual localization setting that requires an underlying SfM model. They thus also provide approaches for keypoint-free SfM. All of them are inherently applicable to object pose estimation (since they do not make any assumption on the type of scenes). Similarly, LoFTR also uses a keypoint-free SfM approach (based on Dual RCNet) (see https://github.com/zju3dv/LoFTR/issues/9). Another approach to keypoint-free SfM based on dense feature matching between images is [Widya et al., Structure from motion using dense CNN features with keypoint relocalization, IPSJ Transactions on Computer Vision and Applications 2018]. Yet, this prior work is not discussed. The differences between this prior work and the proposed approach should be clearly described. Furthermore, comparisons with other keypoint-free approaches, e.g., Patch2Pix or Dual RCNet, are missing.
b) As far as I can see, the coarse reconstruction stage for keypoint-free SfM is the same as for LoFTR (based on the description provided here: https://github.com/zju3dv/LoFTR/issues/9). The refinement stage seems identical to [18]. The paper states that "Note that our keypoint-free SfM framework is also related to PixSfM [18] but comes with different motivations. PixSfM improves keypoint-based SfM for more accurate 3D reconstructions by refining inaccurately-detected sparse local features with dense feature maps. Different from PixSfM, we aim to adapt the keypoint-free method LoFTR [29] to SfM for object pose estimation." However, I disagree with this statement. As stated in the paper, "Every pixel in the downsampled image can be regarded as a “keypoint” in the original image." The goal of the refinement stage is to "refine the object point cloud with sub-pixel correspondences." In other words, the motivation for the refinement stage is to obtain a more accurate 3D model by refining initially inaccurate keypoint positions. This is achieved using dense feature maps to detect more accurate keypoint positions.

W3) As in the case of keypoint-free SfM, there are other directions of highly related work that omitted:
a) Work on object pose estimation using SfM rather than CAD models certainly predates OnePose. Examples include [Gordon & Lowe, What and Where: 3D Object Recognition with Accurate Pose, Toward Category-Level Object Recognition, 2006], [Rothganger et al., 3D Object Modeling and Recognition Using Local Affine-Invariant Image Descriptors and Multi-View Spatial Constraints, 3D Object Modeling and Recognition Using Local Affine-Invariant Image Descriptors and Multi-View Spatial Constraints, IJCV 2006], [Hsiao et al., Making specific features less discriminative to improve point-based 3D object recognition, CVPR 2010[, [Bhat et al., Visual words for 3D reconstruction and pose computation, 3DIM/3DPVT 2011], and [Fenzi et al., 3D Object Recognition and Pose Estimation for Multiple Objects using Multi-Prioritized RANSAC and Model Updating, DAGM/OAGM 2012]. This prior work should be properly acknowledged.
b) The paper states that "HLoc is slow during pose estimation because it depends on multiple 2D-2D image matchings as the proxy for building 2D-3D correspondences." Yet, there is quite some literature on visual localization algorithms that do not use image retrieval but directly match 2D features against 3D points via associated feature descriptors. Examples include: [Arth et al., Wide Area Localization on
Mobile Phones. ISMAR 2009], [Li et al., Location Recognition using Prioritized Feature Matching, ECCV 2010], [Li et al., Worldwide Pose Estimation Using 3D Point Clouds, ECCV 2012], [Choudhary & Narayanan, Visibility probability structure from sfm datasets and applications, ECCV 2012], [Donoser & Schmalstieg, Discriminative featureto-point matching in image-based localization, CVPR 2014], [Cao & Snavely, Minimal scene descriptions from structure from motion models, CVPR 2014], [Lim et al., Real-time monocular image-based 6-dof localization, IJRR 2015], [Lynen et al., Get out of my lab: Largescale, real-time visual-inertial localization, RSS 2015], [Zeisl et al., Camera Pose Voting for Large-Scale Image-Based Localization, ICCV 2015], [Camposeco et al., Toroidal Constraints for TwoPoint Localization under High Outlier Ratios, CVPR 2017], [DuToit et al., Consistent map-based 3d localization on mobile devices, ICRA 2017], [Liu et al., Efficient Global 2D-3D Matching for Camera Localization in a Large-Scale 3D Map, ICCV 2017], [Sattler et al., Efficient & effective prioritized matching for large-scale image-based localization, PAMI 2017], [Svarm et al., City-scale localization for cameras with known vertical direction, PAMI 2017], and [Lynen et al., Large-scale, real-time visual-inertial localization revisited, IJRR 2019]. Many of these approaches are directly applicable to the object pose estimation setting based on SfM models and should thus be discussed.

The following additional comments did not affect my recommendation:
* References for RANSAC and the PnP solver used are missing.

---

> ### Author Response · Authors · 2022-08-02
> **The response for the major concerns (2/2)**
>
> >**Q4:** The relation of OnePose++ to prior work on SfM-based object pose estimation and visual localization.
>
> **R4:** We thank the reviewer for pointing out the missing discussions and the references of the prior works. We describe the relations as follows, and we promise to add them to the final version.
>
> **Discussion with prior works on SfM-based pose estimation**
>
> Compared with previous works on SfM-based pose estimation, our pipeline can be regarded as "renovating the pipeline with a learning-based approach"[30]. The main contribution of our method is the keypoint-free framework to eliminate the pipeline's reliance on detected keypoints. Thus our method achieves improvements on low-textured scenarios, which are challenging for previous methods.
>
> **Discussion with prior works on visual localization**
>
> The previous visual localization methods based on the direct 2D-3D matching focus on handling the large-scale problem, while the main challenge in our task is how to match the query image with the low-textured 3D model for object pose estimation. In our 2D-3D matching network, we eliminate the keypoint detection on the query image and leverage the attention module to provide the global receptive field and yield the discriminative features for 2D-3D matching.
>
> The visualizations in Fig. 5 and discussions in Sec. 4.5 demonstrate the attention module plays a critical role in the 2D-3D matching and pose estimation of low-textured objects.
>
> > **Q5:** References for RANSAC and the PnP solver used are missing.
>
> **R5:** We thank the reviewer for pointing out the missing references. We promise to add them in the final version.

---

> > ### Comment · Reviewer_iDEM · 2022-08-05
> > **Re: The response for the major concerns (2/2)**
> >
> > Thank you very much for the answers. Please find my comments and concerns below.
> >
> > > Compared with previous works on SfM-based pose estimation, our pipeline can be regarded as "renovating the pipeline with a learning-based approach"[30]. The main contribution of our method is the keypoint-free framework to eliminate the pipeline's reliance on detected keypoints. Thus our method achieves improvements on low-textured scenarios, which are challenging for previous methods.
> >
> > This is rather vague to me. What does "renovating the pipeline with a learning-based approach" mean? Prior work, e.g., SuperGlue, LoFTR, Patch2Pix, NC-Net, Sparse NC-Net, Dual RC-Net, DSAC (++, *), InLoc, D2-Net, has already integrated learning into SfM-based pose estimation. Some of these approaches also claim to better handle weakly textured regions (e.g., InLoc motivates dense matching to better handle such regions).
> >
> > > The previous visual localization methods based on the direct 2D-3D matching focus on handling the large-scale problem, while the main challenge in our task is how to match the query image with the low-textured 3D model for object pose estimation. In our 2D-3D matching network, we eliminate the keypoint detection on the query image and leverage the attention module to provide the global receptive field and yield the discriminative features for 2D-3D matching.
> >
> > The challenge at large scale is that feature descriptors become ambiguous as more and more locally similar structures need to be considered (see the work by Li et al., Svarm et al., and Zeisl et al.). The result is that some form of disambiguation is needed, as is the case for weakly texture regions, which also produce ambiguous matches. These works thus need to deal with a very similar problem. In my opinion, the differences need to be discussed in more detail.

---

> > > ### Author Response · Authors · 2022-08-09
> > > **The response to the follow-up questions -4**
> > >
> > > > **Q8:** The challenge at large scale is that feature descriptors become ambiguous as more and more locally similar structures need to be considered (see the work by Li et al., Svarm et al., and Zeisl et al.). The result is that some form of disambiguation is needed, as is the case for weakly texture regions, which also produce ambiguous matches. These works thus need to deal with a very similar problem. In my opinion, the differences need to be discussed in more detail.
> > >
> > > Thank you very much for your comments. Previous visual localization methods based on direct 2D-3D matching improve efficiency, accuracy and cope with ambiguous matches in the 2D-3D matching and outlier filtering.
> > >
> > > Many previous methods [1,2,8] leverage priors for 2D-3D matching. They define the prioritization criteria, such as co-visibility for the 3D points, and matching is performed by order of descending priorities. This strategy improves efficiency but helps little in disambiguation. Some methods [1,4,11] compress the 3D model by quantizing features to improve matching efficiency. However, the quantization can further lead to ambiguous matches and they rely on outlier filtering for disambiguation. [9] regards 2D-3D matching as a classification problem, but it assumes the known pose prior. Our method also works on the 2D-3D matching phase but focuses on disambiguating features. Our idea is to directly disambiguate and augment features by encoding their spatial information and relations with others into features with the help of the attention module. In this way, both 2D and 3D features are provided the global receptive field and become discriminative for 2D-3D matching. Moreover, we eliminate the keypoint detection on the query image, which helps pose estimation of low-textured objects. Since our module operates on the features, we believe it can be combined with the previous prior-based methods to perform 2D-3D matching.
> > >
> > > A number of solutions [1,3,4,5,6,7,10,11] work on the outlier filter stage. Since the ambiguous matches may contain correct matches, some methods relax the matching threshold [3,5,6,7] or quantize features [1,4,11] to preserve ambiguous matches and reject wrong matches at the outlier filter stage. Many approaches[1,4,7,11] use co-visibility priors to filter outliers. The co-visibility encoded in the 3D model is used to select a subset of matches that is more likely to be correct from all putative matches. Some other methods[3,5,6,10] propose efficient geometric verification to filter large amounts of outliers. Since our work focus on the 2D-3D matching phase, these outlier filtering methods are orthogonal to our method, which can be further explored to integrate into our pipeline.
> > >
> > > **References**
> > >
> > > [1] Sattler, Torsten, B. Leibe and Leif P. Kobbelt. “Efficient & Effective Prioritized Matching for Large-Scale Image-Based Localization.” IEEE Transactions on Pattern Analysis and Machine Intelligence 39 (2017): 1744-1756.
> > >
> > > [2] Li, Yunpeng, Noah Snavely and Daniel P. Huttenlocher. “Location Recognition Using Prioritized Feature Matching.” ECCV (2010).
> > >
> > > [3] Svärm, Linus, Olof Enqvist, Fredrik Kahl and Magnus Oskarsson. “City-Scale Localization for Cameras with Known Vertical Direction.” IEEE Transactions on Pattern Analysis and Machine Intelligence 39 (2017): 1455-1461.
> > >
> > > [4] Liu, Liu, Hongdong Li and Yuchao Dai. “Efficient Global 2D-3D Matching for Camera Localization in a Large-Scale 3D Map.” 2017 IEEE International Conference on Computer Vision (ICCV) (2017): 2391-2400.
> > >
> > > [5] Camposeco, Federico, Torsten Sattler, Andrea Cohen, Andreas Geiger and Marc Pollefeys. “Toroidal Constraints for Two-Point Localization Under High Outlier Ratios.” 2017 IEEE Conference on Computer Vision and Pattern Recognition (CVPR) (2017): 6700-6708.
> > >
> > > [6] Zeisl, Bernhard, Torsten Sattler and Marc Pollefeys. “Camera Pose Voting for Large-Scale Image-Based Localization.” 2015 IEEE International Conference on Computer Vision (ICCV) (2015): 2704-2712.
> > >
> > > [7] Li, Yunpeng, Noah Snavely, Daniel P. Huttenlocher and Pascal V. Fua. “Worldwide Pose Estimation Using 3D Point Clouds.” ECCV (2012).
> > >
> > > [8] Choudhary, Siddharth and P. J. Narayanan. “Visibility Probability Structure from SfM Datasets and Applications.” ECCV (2012).
> > >
> > > [9] Donoser, Michael and Dieter Schmalstieg. “Discriminative Feature-to-Point Matching in Image-Based Localization.” 2014 IEEE Conference on Computer Vision and Pattern Recognition (2014): 516-523.
> > >
> > > [10] Svärm, Linus, Olof Enqvist, Magnus Oskarsson and Fredrik Kahl. “Accurate Localization and Pose Estimation for Large 3D Models.” 2014 IEEE Conference on Computer Vision and Pattern Recognition (2014): 532-539.
> > >
> > > [11] Sattler, Torsten, Michal Havlena, Filip Radenović, Konrad Schindler and Marc Pollefeys. “Hyperpoints and Fine Vocabularies for Large-Scale Location Recognition.” 2015 IEEE International Conference on Computer Vision (ICCV) (2015): 2102-2110.

---

> > > > ### Comment · Reviewer_iDEM · 2022-08-10
> > > > **Re: The response to the follow-up questions -4**
> > > >
> > > > Thank you very much for the detailed description of the relation to prior work.
> > > >
> > > > > Our idea is to directly disambiguate and augment features by encoding their spatial information and relations with others into features with the help of the attention module.
> > > >
> > > > There is actually prior work on using 3D point relations (in terms of which points are covisible with each other) to disambiguate matches, e.g., see Sec. 4 in [11]. In essence, image retrieval / determining which matching points can be seen together encodes relations between 3D points for disambiguation.
> > > >
> > > > > Moreover, we eliminate the keypoint detection on the query image, which helps pose estimation of low-textured objects.
> > > >
> > > > There is prior work on eliminating keypoint detection in the query image for the purpose of better handling challenging conditions where features cannot be reliably re-detected. For example, [Germain et al., Sparse-to-Dense Hypercolumn Matching for Long-Term Visual Localization, 3DV 2019] and [Germain et al., S2DNet : Learning Image Features for Accurate Sparse-to-Dense Matching, ECCV 2020] match sparse features extracted from database images against dense features extracted from a query image.

---

> > > ### Author Response · Authors · 2022-08-09
> > > **The response to the follow-up questions -3**
> > >
> > > > **Q7:** This is rather vague to me. What does "renovating the pipeline with a learning-based approach" mean? Prior work, e.g., SuperGlue, LoFTR, Patch2Pix, NC-Net, Sparse NC-Net, Dual RC-Net, DSAC (++, *), InLoc, D2-Net, has already integrated learning into SfM-based pose estimation. Some of these approaches also claim to better handle weakly textured regions (e.g., InLoc motivates dense matching to better handle such regions).
> > >
> > > Thank you very much for your comments. We clarify that our comment "renovating the pipeline with a learning-based approach" refers to the comparison with the previous SfM-based methods[1,2,3,4,5] in the area of 6D object pose estimation. We further discuss the differences with these methods as follows.
> > >
> > > Some previous methods[2,3,5] extract keypoints on the query image firstly and perform matching with reference images or SfM model to obtain 2D-3D matches for pose estimation. Unlike them, which reject ambiguous matches by ratio test in matching, [4] proposes preserving ambiguous matches at the matching stage by vector quantizing and solving ambiguation by hypothesis testing at the outlier filter stage. [1] proposes the spatial feature clustering and multi-prioritized RANSAC to cope with repeated patterns for multiple instances detection.
> > >
> > > Different from these previous methods, our framework eliminates the keypoint detection for the query image by directly performing matching between the 2D feature map and the 3D model, which benefits pose estimation for low-textured objects. Moreover, we leverage the attention mechanism to disambigute 2D and 3D features for matching, while the direct feature disambiguation is not explored by these methods. The keypoint-free design and the attention mechansim in our 2D-3D matching network bring improvement on low-textured objects, which are challenging for these previous methods.
> > >
> > > **References:**
> > >
> > > [1] Fenzi, Michele, Ralf Dragon, Laura Leal-Taixé, Bodo Rosenhahn and Jörn Ostermann. “3D Object Recognition and Pose Estimation for Multiple Objects Using Multi-Prioritized RANSAC and Model Updating.” DAGM/OAGM Symposium (2012).
> > >
> > > [2] Martinez, Manuel, Alvaro Collet and Siddhartha S. Srinivasa. “MOPED: A scalable and low latency object recognition and pose estimation system.” 2010 IEEE International Conference on Robotics and Automation (2010): 2043-2049.
> > >
> > > [3] Gordon, Iryna and David G. Lowe. “What and Where: 3D Object Recognition with Accurate Pose.” Toward Category-Level Object Recognition (2006).
> > >
> > > [4] Hsiao, Edward, Alvaro Collet and Martial Hebert. “Making specific features less discriminative to improve point-based 3D object recognition.” 2010 IEEE Computer Society Conference on Computer Vision and Pattern Recognition (2010): 2653-2660.
> > >
> > > [5] Skrypnyk, Iryna and David G. Lowe. “Scene modelling, recognition and tracking with invariant image features.” Third IEEE and ACM International Symposium on Mixed and Augmented Reality (2004): 110-119.

---

> > > > ### Comment · Reviewer_iDEM · 2022-08-10
> > > > **Re: The response to the follow-up questions -3**
> > > >
> > > > > Thank you very much for your comments. We clarify that our comment "renovating the pipeline with a learning-based approach" refers to the comparison with the previous SfM-based methods[1,2,3,4,5] in the area of 6D object pose estimation.
> > > >
> > > > Thanks for the clarification. I misunderstand the previous statement as something more general pertaining to SfM-based pose estimation in general, which would also include the visual localization literature (where learning by now is a central part of the pipeline).
> > > >
> > > > Still, it might be better to not make this statement, as the approaches from the localization literature are also applicable to the object pose estimation case (see your experiments shown above). As such, claiming to renovate the pipeline with learning-based approaches seems still to strong to me.

---

> ### Author Response · Authors · 2022-08-02
> **The response for the major concerns (1/2)**
>
> We thank the reviewers for the insightful suggestions. We address the major concerns below:
> >**Q1:** Both OnePose and OnePose++ are misclassified as being generalizable object pose estimation approaches. It is not too surprising that it outperforms the generalizable baselines as it is able to train (in the form of building a SfM model) per object.
>
> **R1:** Thank you very much for your comments. We clarify that the 'generalizable' baselines in our paper include OnePose, HLoc, Gen6D, which are given the same input and share exactly the same setting as ours. Therefore, the comparison in our experiments is fair and substantial. We follow the naming of previous methods OnePose[30] and Gen6D[19], which denote the property of eliminating object/category-specific training as ‘generalizability’.
>
> >**Q2:** Please describe in detail how the proposed keypoint-free SfM approach differs from prior work in this area.
>
> **R2:** We clarify the difference with previous works as follows and promise to add the discussion and reference of these previous methods in the final version.
>
> **Compare with PixSfM[18]:**
> The main difference in refinement is that we leverage fine-level matching with Transformer to refine the 2D locations of coarse feature tracks and then optimize the 3D model with geometric error, while PixSfM uses pre-stored dense feature maps and feature-metric BA to refine the 3D model and 2D keypoints globally.
> The advantages of our refinement are
>
> - Accuracy. The capability of the two-view transformer module in fine-level matching can be leveraged by our refinement to find accurate matches at low-texture regions, where the CNN feature map used by PixSfM struggles.
> - Storage Efficiency. We do not need to extract and store dense local features around each 2D point and keep them in memory to perform feature-metric optimization like PixSfM. Therefore the storage and memory peak during refinement is low.
>
> We report the point cloud accuracy evaluated on OnePose-HARD scanned objects as follows. The results demonstrate that the 3D models reconstructed by our refinement achieve higher accuracy. Our refinement is also more storage efficient in terms of dense features storage cost. Notably, the image resolution in the dataset is 512×512. With image resolution increase, the storage cost of PixSfM will rise significantly since keypoint-free matchers will yield much more matches.
> ||1mm|3mm|5mm|Feature Storage Cost|
> |:-|:-|:-|:-|:-|
> |LoFTR coarse + Our refinement|**29.5**|**73.6**|**85.8**|-|
> |LoFTR coarse + PixSfM|27.6|71.2|84.4|7.35GB|
> |LoFTR coarse (no refinement)|25.6|68.9|83.6|-|
>
> The following results evaluated on the OnePose dataset illustrate that our refinement also brings improvement for the object pose estimation.
> ||1cm1deg|3cm3deg|5cm5deg|
> |:-|:-|:-|:-|
> |LoFTR coarse + Our refinement|**50.7**|**80.0**|**87.0**|
> |LoFTR coarse + PixSfM|48.9|79.3|86.4|
> |LoFTR coarse (no refinement)|45.5|78.6|86.0|
>
> **Compare with SfM approaches provided by keypoint-free descriptors:**
>
> The main difference is that all these approaches face the trade-off between point accuracy and repeatability. I.e., they scarface the sub-pixel match accuracy by rounding matches to grid level or merging matches within a grid to obtain repeatable 'keypoints' for SfM. On the contrary, our SfM obtains repeatable features while preserving the sub-pixel matching accuracy by the refinement phase.
>
> **Compare with [Widya et al., Structure from motion using dense CNN features with keypoint relocalization]:**
>
> [Widya et al.] and OnePose++ share a similar pipeline in terms of SfM, which firstly strikes for repeatable matches with low-res dense feature grids, then refines matching positions for higher accuracy.
>
> The main difference in refinement is that [Widya et al.] only leverages the local information from each matched point to relocalize points. Since the lack of two-view or multiview constraints, the keypoint detection noise exists in its relocalization phase.
> In contrast, our refinement phase performs multiple two-view dense matching in the local regions based on the coarse feature tracks to refine 2D point locations. Therefore, the detection error is avoided, and the capability of keypoint-free matchers' fine level matching is leveraged to boost the performance on low-textured objects.
>
> >**Q3:** Why were Patch2Pix, Dual RCNet, etc., not considered as baselines?
>
> **R3:** The main reason is that these methods are not state-of-the-art regarding their performance on both two-view matching and visual localization. We also conduct experiments on the OnePose-HARD dataset to compare our pipeline with these methods based on their SfM and localization methods. The results show that our method outperforms them by a large margin. We will add the results and references in the final version.
>
> ||1cm1deg|3cm3deg|5cm5deg|
> |:-|:-|:-|:-|
> |Ours|**16.3**|**55.4**|**70.3**|
> |LoFTR(round)|15.4|43.7|53.4|
> |DRC-Net|11.3|37.0|47.8|
> |Patch2Pix|2.42|19.0|30.4|

---

> > ### Comment · Reviewer_iDEM · 2022-08-05
> > **Re: The Response for the major concerns (1/2) - 1**
> >
> > Thank you very much for the detailed feedback. Please find my comments and follow-up questions below.
> >
> > > **R1**: Thank you very much for your comments. We clarify that the 'generalizable' baselines in our paper include OnePose, HLoc, Gen6D, which are given the same input and share exactly the same setting as ours. Therefore, the comparison in our experiments is fair and substantial. We follow the naming of previous methods OnePose[30] and Gen6D[19], which denote the property of eliminating object/category-specific training as ‘generalizability’.
> >
> > If generalizability is defined as "the property of eliminating object/category-specific training", then I don't think that OnePose, HLoc, and OnePose++ are qualify as generalizable. They all need to build an object/category-specific scene representation, in the form of a 3D model, from the input images and their known poses. I don't why building these 3D models would not qualify as object/category-specific training as it involves optimizing an objective function and since these 3D models are fundamental parts of the object pose estimation stage.
> >
> > > **R2**: We clarify the difference with previous works as follows and promise to add the discussion and reference of these previous methods in the final version.
> >
> >
> > **Relation to [Dusmanu et al., Multi-View Optimization of Local Feature Geometry, ECCV 2020]**
> >
> > The closer I look at the proposed refinement, the more it seems a special case of the approach of Dusmanu et al. In their work, Dusmanu et al. deal with refining keypoint positions for feature matches. For the two-view case, they estimate the flow from one keypoint to a position in a patch around the other matching keypoint by matching features and regressing the flow. This seems conceptually the same as the fine matching stage (with probably the main difference being that Dusmanu et al. did not use a transformer). For the multi-view case, Dusmanu et al. state that "Firstly, since corresponding features are generally observed from different viewpoints and looking at non-planar scene structures, the computed displacement vector is only valid for the central pixel and not constant within the patch [...]. Thus, when refining keypoint locations u, v, w, . . . over multiple views, consistent results can only be produced by forming displacement chains (e.g., du→v + d(v+du→v)→w + . . .) without loops. However, such an approach does not consider all possible edges in the graph and quickly accumulate errors along the chain." Rather than computing the offset / flow for a single pixel (the original feature position) to a patch, Dusmanu et al. thus compute flow fields between patches and use these fields to jointly refine all keypoint positions (after fixing one keypoint position in one of the images), using as many pairwise matches as possible. The proposed approach is a special case in the sense that (1) it uses a sub-graph of the pairwise matching graph that connects the reference node with the other nodes, but does not include any connections between the other nodes, and (2) only computes a single offset per pair and not a full flow field.
> >
> > Unless I am overlooking something, I believe that the claim that a novel keypoint-less SfM approach is proposed needs to be adjusted.
> >
> > **Compare with PixSfM[18]:**
> >
> > Thank you very much for the detailed answer. I have two comments / questions:
> >
> > > Accuracy. The capability of the two-view transformer module in fine-level matching can be leveraged by our refinement to find accurate matches at low-texture regions, where the CNN feature map used by PixSfM struggles.
> >
> > Where can I see that PixSfM struggles in "low-texture regions"? The reported performance seems rather very similar to me.
> >
> > > Storage Efficiency. We do not need to extract and store dense local features around each 2D point and keep them in memory to perform feature-metric optimization like PixSfM. Therefore the storage and memory peak during refinement is low.
> >
> > Looking at Sec. 4.4 of the PixSfM paper (and its supplementary material), the high memory costs can be avoided by pre-computing and storing cost maps rather than the descriptors. This comes at a small loss in accuracy. In the provided table, are the 7.35GB required for storing descriptors or the cost maps?

---

> > > ### Comment · Reviewer_iDEM · 2022-08-05
> > > **Re: The Response for the major concerns (1/2) - 2**
> > >
> > > > The main difference in refinement is that [Widya et al.] only leverages the local information from each matched point to relocalize points. Since the lack of two-view or multiview constraints, the keypoint detection noise exists in its relocalization phase.
> > >
> > > Widya et al. start with coarse matches that are then refined locally: given a match established using features extracted at one layer in the network, the refinement aims at finding more accurate coordinates locally in regions around the initial match (where the region size depends on the receptive field of the features). Isn't this a similar two-view constraint used by the proposed approach?
> > >
> > > > **R3**: The main reason is that these methods are not state-of-the-art regarding their performance on both two-view matching and visual localization.
> > >
> > > Using Patch2Pix to refine matches found by SuperGlue (denoted as SuperGlue + Patch2Pix in the Patch2Pix paper) leads to state of the art results for the visual localization task (results from visuallocalization.net) (higher is better):
> > >
> > > | Method | Aachen Day-Night v1.1 | InLoc |
> > > |----------|----------------------------|--------|
> > > | LoFTR  | day: 88.7 / 95.6 / 99.0, night: 78.5 / 90.6 / 99.0 | duc1: 47.5 / 72.2 / 84.8, duc2: 54.2 / 74.8 / 85.5 |
> > > | SuperGlue + Patch2Pix | day: 89.3 / 95.8 / 99.2, night: 78.0 / 90.6 / 99.0 | duc1: 50.0 / 68.2 / 81.8, duc2: 57.3 / 77.9 / 80.2 |
> > > | Patch2Pix | day: 86.4 / 93.0 / 97.5, night: 72.3 / 88.5 / 97.9 | duc1: 44.4 / 66.7 / 78.3, duc2: 49.6 / 64.9 / 72.5 |
> > >
> > > Unfortunately, this stronger baseline is missing.

---

> > > > ### Author Response · Authors · 2022-08-09
> > > > **The response to the follow-up questions -2**
> > > >
> > > > > **Q5:** Widya et al. start with coarse matches that are then refined locally: given a match established using features extracted at one layer in the network, the refinement aims at finding more accurate coordinates locally in regions around the initial match (where the region size depends on the receptive field of the features). Isn't this a similar two-view constraint used by the proposed approach?
> > > >
> > > > Thank you very much for your comments. The keypoint relocalization phase in Widya et al. doesn't leverage the two-view constraint. In its refinement phase, the keypoint relocalization still operates on a single view by the local feature patch instead of considering relations with other views' patches. It can be regarded as performing keypoint detection within the coarse match local region. In contrast, our refinement leverage two-view patches and transformer to find more accurate matches in the query view relative to the reference view.
> > > >
> > > > > **Q6:** Using Patch2Pix to refine matches found by SuperGlue (denoted as SuperGlue + Patch2Pix in the Patch2Pix paper) leads to state of the art results for the visual localization task. Unfortunately, this stronger baseline is missing.
> > > >
> > > > Thank you very much for your comments. Since the SuperGlue+Patch2Pix is not a keypoint-free matcher, we did not include this baseline in the answer of the original Q3. We apologize for misunderstanding the question, and we add the evaluation of SuperGlue+Patch2Pix on the OnePose-HARD dataset as follows.
> > > >
> > > > ||1cm1deg|3cm3deg|5cm5deg|
> > > > |:-|:-|:-|:-|
> > > > |Ours|**16.3**|**55.4**|**70.3**|
> > > > |LoFTR(round)|15.4|43.7|53.4|
> > > > |SPP+SPG+Patch2Pix|10.1|37.2|47.6|
> > > > |SPP+SPG|13.8|36.1|42.2|
> > > > |DRC-Net|11.3|37.0|47.8|
> > > > |Patch2Pix|2.42|19.0|30.4|

---

> > > > > ### Comment · Reviewer_iDEM · 2022-08-10
> > > > > **Re: The response to the follow-up questions -2**
> > > > >
> > > > > >  The keypoint relocalization phase in Widya et al. doesn't leverage the two-view constraint.
> > > > >
> > > > > You are right. I seem to have confused things with the two-view-based refinement from the InLoc paper [Taira et al., CVPR 2018], which implements the refinement strategy based on local matching discussed in the beginning of Sec. 3.3 of Widya et al.
> > > > >
> > > > > > SuperGlue+Patch2Pix is not a keypoint-free matcher
> > > > >
> > > > > Thank you very much for the additional results. I think this rounds out the experiment.
> > > > >
> > > > > I would however argue that SuperGlue+Patch2Pix is a keypoint-free matcher as the Patch2Pix stage refines the SuperGlue matches in terms of their spatial positions. As a result, the matching pixel positions can differ from the original feature detections.

---

> > > ### Author Response · Authors · 2022-08-09
> > > **The response to the follow-up questions -1**
> > >
> > > > **Q1:** If generalizability is defined as "the property of eliminating object/category-specific training", then I don't think that OnePose, HLoc, and OnePose++ are qualify as generalizable. They all need to build an object/category-specific scene representation, in the form of a 3D model, from the input images and their known poses. I don't why building these 3D models would not qualify as object/category-specific training as it involves optimizing an objective function and since these 3D models are fundamental parts of the object pose estimation stage.
> > >
> > > Thank you very much for your comments. We will change 'generalizability' to 'no object-specific network training' in the revised version to avoid misleading.
> > >
> > > > **Q2:** Relation to PatchFlow[Dusmanu et al., Multi-View Optimization of Local Feature Geometry, ECCV 2020]
> > >
> > > Thank you very much for your comments. The differences between PatchFlow are as follows.
> > >
> > > - We leverage the fine-level matching module with the transformer to refine matches instead of estimating the dense flow field between patches like PatchFlow. The advantages are accuracy and storage efficiency. Since the multiview refinement of PatchFlow requires flow field interpolation, the fine matching module is not adaptable for its framework.
> > > - We thus propose a simple yet effective strategy to achieve consistent matches for later 3D model refinement. We keep the selected reference node fixed and search around each query node for the fine-level match. The advantages are that our graph structure is significantly simpler than PatchFlow, which is efficient for matching. And we do not need to store and interpolate flow fields for optimization.
> > >
> > > Notably, our graph structure is not a sub-graph of the coarse feature track since it contains connections which not exist in the coarse feature track(e.g., the reference and the query node may not be directly matched in tentative matches).
> > >
> > > > **Q3:** Where can I see that PixSfM struggles in "low-texture regions"? The reported performance seems rather very similar to me.
> > >
> > > The feature cost maps of the local patch around the coarse matches are visualized [here](https://sites.google.com/view/oneposeplusplus/%E9%A6%96%E9%A1%B5). The feature cost maps are calculated by the distance between the corresponding reference feature of another view and each element within the local patch, and the values are normalized to 0~1 for visualization.
> > >
> > > It can be seen that the feature distance maps of PixSfM contain large or multiple minimal regions(blue region) incurred by the ambiguous CNN features in low-textured regions, which are not discriminative enough to find real optimal locations in feature-metric optimization.  We attribute the accuracy improvement of our method to the discriminative features in fine-level matching.
> > >
> > > > **Q4:** Looking at Sec. 4.4 of the PixSfM paper (and its supplementary material), the high memory costs can be avoided by pre-computing and storing cost maps rather than the descriptors. This comes at a small loss in accuracy. In the provided table, are the 7.35GB required for storing descriptors or the cost maps?
> > >
> > > The reported memory requirement is for storing descriptors. We also conduct the experiment for PixSfM with cost maps on the OnePose-HARD dataset. The results show that although the feature storage cost decreases significantly, the accuracy decreases accordingly.
> > > ||1mm|3mm|5mm|Feature Storage Cost|
> > > |:-|:-|:-|:-|:-|
> > > |LoFTR coarse + Our refinement|**29.5**|**73.6**|**85.8**|-|
> > > |LoFTR coarse + PixSfM|27.6|71.2|84.4|7.35GB|
> > > |LoFTR coarse + PixSfM(cost map)|25.8|67.4|80.8|0.17GB|

---

> > > > ### Comment · Reviewer_iDEM · 2022-08-09
> > > > **Re: The response to the follow-up questions -1**
> > > >
> > > > Thank you very much for the detailed answers. Please find my comments below.
> > > >
> > > > > Thank you very much for your comments. We will change 'generalizability' to 'no object-specific network training' in the revised version to avoid misleading.
> > > >
> > > > This is indeed a better description of OnePose, OnePose++, hloc, etc.
> > > >
> > > > Note that not requiring "object-specific network training" is not a virtue in itself. For example, the baseline using NeuS in the discussion above does not require object-specific network training. But given the long reconstruction times, it would certainly be feasible to do object-specific network training in the same time. I don't think it matters whether an instance-level method requires network training or not. The important question to me seems to be the trade-off between pose accuracy and the time required to adapt an approach to a given object instance (e.g., training network parameters or building a 3D model).
> > > >
> > > > > The differences between PatchFlow are as follows.
> > > >
> > > > Based on the description, I would still argue that the proposed approach is a special case of PatchFlow:
> > > >
> > > > 1. Refining matches between pairs of matches is a special case of computing a flow field (as only the flow for a single pixel in the patch around one match is computed) and is conceptually identical to the two-view case of PatchFlow (with the main technical difference that a transformer is used, but I would not consider this too novel).
> > > > 2. "We keep the selected reference node fixed and search around each query node for the fine-level match.": In essence, this corresponds to the chaining of refined matches discussed in the PatchFlow paper, with the special structure that the resulting graph has a star-like structure. Since potential constraints between query nodes are not taken into account, the resulting graph is a sub-graph of the one used by PatchFlow. Hence, the proposed approach is a special case of the more general PatchFlow framework.
> > > >
> > > > Thus, I see limited novelty in this part of the proposed approach.
> > > >
> > > > > It can be seen that the feature distance maps of PixSfM contain large or multiple minimal regions(blue region) incurred by the ambiguous CNN features in low-textured regions, which are not discriminative enough to find real optimal locations in feature-metric optimization. We attribute the accuracy improvement of our method to the discriminative features in fine-level matching.
> > > >
> > > > My point was that based on the numbers provided above (point cloud and object pose accuracy), the difference between the proposed approach and PixSfM seems to be very small. The new visualizations indeed show a difference, but these do not seem to influence the quantitative results too much. Hence, the statement that PixSfM struggles with low-texture regions seems to be too strong given the similar numbers.
> > > >
> > > > **Q4**: Thank you very much, this answers my question.

---

### Official Review · Reviewer_hB2N · 2022-07-11

**Rating:** 5
**Confidence:** 3
**Soundness:** 2 fair
**Presentation:** 3 good
**Contribution:** 2 fair

**Summary:**

The paper looks at the problem of one-shot object pose estimation on textureless objects where previous keypoint-based methods fail to perform well. The main contribution is using the keypoint-free SFM pipeline to create a repeatable semi-dense point cloud, which automatically helps improve 3D-2D correspondence to estimate the object pose. Comparisons show that the method performs better than a previous method which uses sparse point cloud reconstructions.

**Questions:**

How is the current method's dense reconstruction different from Multi-View Stereo paradigms. Why are evaluations not compared to these paradigms to show the accuracy improvement if the dense reconstruction is very good?



**Limitations:**

The limitations and potential negative impact are well studied in the paper.

**Strengths And Weaknesses:**

Strengths:
- Moving away from keypoint-based methods helps in automatic object pose estimation of objects in the wild.
- The semi-dense reconstruction of the objects seems to be very helpful in the pose estimation due to better 2D-3D correpondences.
- The results show that the method is more robust to occlusions than previous methods and a live demo helps show the method's accuracy.
- Ablation study shows the advantages of the refinement step and the attention module in the pose estimation network.
- Openpose-Hard dataset is useful for research in pose estimation of textureless objects.

Weaknesses:
- The major advantage or the difference between the proposed network and the previous baseline (onepose [30]) is the dense object reconstruction of the objects from videos. The contribution in the dense reconstruction from videos is well studied in the Multi-View Stereo frameworks. So I feel that the method doesn't have a lot of novelty in terms of reconstruction.
- The paper assumes that the object videos are given aprior, so one-shot object pose estimation might be a misleading term as the method will fail if the objects videos are not provided beforehand.
-  Literature review of Multi-view stereo needs to be well studied and the difference between these methods and the proposed keypoint-free methods need to be well established.
- The evaluations are not substantial as comparisons to other methods like CAD-model based pose estimation have not been well studied.

---

> ### Author Response · Authors · 2022-08-02
> **The response for the major concerns**
>
> We thank the reviewers for the insightful suggestions. We address the major concerns below:
> >**Q1:** How is the current method's dense reconstruction different from Multi-View Stereo paradigms? The major advantage or the difference between the proposed network and the previous baseline (onepose [30]) is the dense object reconstruction of the objects from videos. The contribution in the dense reconstruction from videos is well studied in the Multi-View Stereo frameworks. So I feel that the method doesn't have a lot of novelty in terms of reconstruction.
>
> **R1:** The comment 'the current method's dense reconstruction' may be a misunderstanding. As described in the paper, our reconstruction part is still an SfM-based method instead of Multi-View Stereo based, and we denote our SfM point cloud as semi-dense since we adapt the keypoint-free image matcher LoFTR, which performs semi-dense matching, to the SfM framework. Therefore our SfM-based pipeline should still be categorized into sparse reconstruction methods, and the reconstructed point cloud is significantly sparser than the dense reconstruction since we do not perform pixel-wise depth estimation such as PatchMatch or PlaneSweep.
>
> Compared with OnePose, our contribution in the reconstruction part is our SfM design to adapt keypoint-free feature matching methods to the SfM. As discussed in L38-42, the keypoint-free matcher LoFTR cannot be directly used for SfM since the inconsistent matches. Our keypoint-free SfM framework solves this problem and yields more complete 3D point clouds compared with the previous keypoint-based SfM framework, which benefits pose estimation.
>
> >**Q2:** Why are evaluations not compared to these paradigms to show the accuracy improvement if the dense reconstruction is very good?
>
> **R2:** The main reason is that there is no existing baseline that performs dense reconstruction on the given video and estimates object poses without object-specific training, i.e., identical to our setting.
>
> Since our setting aims for efficient pose estimation with the given video, we believe the SfM-based sparse reconstruction is more suitable for the setting because it is more computationally efficient than dense reconstruction.
>
> Moreover, the reconstructed SfM point clouds are more compact than the dense MVS point clouds because they are sparser and contain mainly informative 3D points, thus more suitable for storing 3D point features and efficient for performing direct 2D-3D matching in our pipeline.
>
> We believe incorporating the dense reconstruction methods for object pose estimation in our setting can be explored as a direction for future works.
>
> >**Q3:** The paper assumes that the object videos are given aprior, so one-shot object pose estimation might be a misleading term as the method will fail if the objects videos are not provided beforehand.
>
> **R3:** Thank you very much for your comments. We clarify that the 'one-shot' naming indicates the setting that given one video shot of the object with annotated poses, our method can estimate its poses in arbitrary environments without additional pose estimator training.
>
> This is similar to the "one-shot" setting in 2D detection and segmentation [1,2,3], which assumes "given an example image of a novel, previously unknown object category (the reference), find and segment all objects of this category within a complex scene (the query image)"[2]
>
> >**Q4:** Literature review of Multi-view stereo needs to be well studied and the difference between these methods and the proposed keypoint-free methods need to be well established.
>
> **R4:** Thank you very much for your comments. We will add the review of Multi-View Stereo methods and the discussion with our keypoint-free SfM framework in the final version.
>
> >**Q5:** The evaluations are not substantial as comparisons to other methods like CAD-model based pose estimation have not been well studied.
>
> **R5:** The comment "comparisons to other methods like CAD-model based pose estimation have not been well studied" may be a misunderstanding. In fact, we compare the proposed method with CAD-model-based baselines PVNet and CDPN on multiple datasets, as pointed out in Line 16-17, shown in Tab 2, 3 and discussed in Sec 4.3, 4.4.
>
> The results demonstrate that our method achieves comparable results with CAD-model-based pose estimation methods, which are trained for each object with the given CAD model.
>
> **References**
>
> [1] Li, Xiang, Lin Zhang, Yau Pun Chen, Yu-Wing Tai and Chi-Keung Tang. “One-Shot Object Detection without Fine-Tuning.” ArXiv abs/2005.03819 (2020): n. pag.
>
> [2] Michaelis, Claudio, Ivan Ustyuzhaninov, Matthias Bethge and Alexander S. Ecker. “One-Shot Instance Segmentation.” ArXiv abs/1811.11507 (2018): n. pag.
>
> [3] Caelles, Sergi, Kevis-Kokitsi Maninis, Jordi Pont-Tuset, Laura Leal-Taixé, Daniel Cremers and Luc Van Gool. “One-Shot Video Object Segmentation.” 2017 IEEE Conference on Computer Vision and Pattern Recognition (CVPR) (2017): 5320-5329.

---

> > ### Comment · Reviewer_iDEM · 2022-08-05
> > **Re: The response for the major concerns**
> >
> > Thank you very much for the detailed answer. I have some follow-up questions and comments regarding **R2**:
> >
> > > The main reason is that there is no existing baseline that performs dense reconstruction on the given video and estimates object poses without object-specific training, i.e., identical to our setting.
> >
> > Wouldn't the following be a suitable (and rather simple) baseline?
> > * At training time, create a dense 3D model of the object, e.g., using MVS.
> > * At test time:
> >   * Match features between the query image and the training images (as is done, e.g., by hloc) to obtain 2D-2D matches.
> >   * Rather than obtaining 2D-3D matches using 3D points (from SfM) associated with the features in the training images, corresponding 3D points can be obtained by rendering depth maps of the dense model (the same is done by localization methods evaluating on the InLoc dataset).
> >   * Do pose estimation with all 2D-3D matches.
> >
> > > Since our setting aims for efficient pose estimation with the given video, we believe the SfM-based sparse reconstruction is more suitable for the setting because it is more computationally efficient than dense reconstruction.
> >
> > I am not sure I understand why this argument holds. After all, the proposed approach is based on dense matching between images, as is the case for dense MVS. I don't see why dense MVS would thus be necessarily faster. E.g., according to the supp. mat., the proposed SfM method takes 347 seconds for 193 images at a resolution of 512x512 pixels. For comparison, starting with known extrinsics and intrinsics, Reality Capture, a state-of-the-art commercial 3D reconstruction system, takes 30 seconds to build a sparse point cloud from 392 images at size 800x600 for scan 65 of the DTU dataset (including feature extraction and matching). Dense reconstruction, including generating a mesh, then takes 394 seconds and computing per-vertex colors for the mesh takes about another 30 seconds. This shows that dense reconstruction is feasible in a comparable time.

---

> > > ### Author Response · Authors · 2022-08-09
> > > **The response to the follow-up questions**
> > >
> > > > **Q1:** Wouldn't the following be a suitable (and rather simple) baseline?...
> > >
> > > Thank you very much for your comments. We observe the mentioned pipeline is similar to MeshLoc[1], which is a recent concurrent work. We follow the mentioned pipeline to conduct the evaluation on the OnePose-HARD dataset.
> > >
> > > We use the current state-of-the-art object dense reconstruction method NeuS[2] to reconstruct object mesh for each object in the dataset, which takes ~10 hours per object. Then we render depth maps for reference images and estimate the object pose of the query image following the mentioned pipeline.
> > > Results are shown as follows.
> > > ||1cm1deg|3cm3deg|5cm5deg| Reconstruction Time (per object) | Pose Estimation Time (per frame)|
> > > |:-|:-|:-|:-|:-|:-|
> > > |Ours|**16.3**|**55.4**|**70.3**| **347s**|**87ms**|
> > > |Neus + LoFTR|15.5|49.9|61.8| ~10 hours| 897ms|
> > > |Neus + Patch2Pix(SPG)|12.5|43.7|55.0|~10 hours| 936ms|
> > >
> > > The results demonstrate that our method achieves higher accuracy, and both the reconstruction and pose estimation are significantly faster.
> > >
> > > **Reference:**
> > >
> > > [1] Pánek, Vojtěch, Zuzana Kukelova and Torsten Sattler. “MeshLoc: Mesh-Based Visual Localization.” ArXiv abs/2207.10762 (2022): n. pag.
> > >
> > > [2] Wang, Peng, Lingjie Liu, Yuan Liu, Christian Theobalt, Taku Komura and Wenping Wang. “NeuS: Learning Neural Implicit Surfaces by Volume Rendering for Multi-view Reconstruction.” NeurIPS (2021).

---

> > > > ### Comment · Reviewer_iDEM · 2022-08-09
> > > > **Re: The response to the follow-up questions**
> > > >
> > > > Thank you very much for the answer and the additional experiment.
> > > >
> > > > I would not count this baseline as concurrent work. The idea of getting 3D point coordinates from a dense model instead of a sparse SfM point cloud predates the MeshLoc paper (I think the MeshLoc paper provides multiple references to such prior work). One example are methods evaluated on the InLoc dataset, where database images are very sparse and building a SfM model is thus hard. Since the dataset provides a depth map per database image, these depth maps are used to obtain the 3D points corresponding to 2D positions in the database images. The methods that I am aware of that evaluate on InLoc rely on image retrieval and run pose estimation separately for the 2D-3D matches obtained from a retrieved database image. Running a single pose estimation step over all 2D-3D matches seems like a minor modification to me.
> > > >
> > > > I am not convinced that the results show that the baseline is not feasible in practice. As mentioned in my previous comment, there are multi-view stereo approaches that are very efficient, e.g., Capturing Reality. NeuS does not seem to fall into this category.

---

### Official Review · Reviewer_BWks · 2022-07-11

**Rating:** 6
**Confidence:** 4
**Soundness:** 3 good
**Presentation:** 3 good
**Contribution:** 3 good

**Summary:**

The authors proposed 6-DoF object pose estimation algorithms which does not require CAD models of target objects.  They require only video sequence of the target object with camera pose.  The keypoint-free SfM build 3D models in training and the keypoint-free 2D-3D matching network can estimate the correspondences between the models and image query. The proposed method handles low-textured objects and achieved state-of-the-art accuracy on public dataset.

**Questions:**

- How about processing time for training and testing?
- Is the pose estimation network trained for each object?

**Limitations:**

- CAD model or its equivalent can be reconstructed from the movie of target object with surrounded AR markers.  This might decrease the advantage of the proposed method.

**Strengths And Weaknesses:**

Strengths:
- The proposed algorithm can handle low-textured objects and does not require CAD model for training, those are useful for real applications.
- The ablation study shows the effectiveness of each component.

Weaknesses:
- The proposed method needs off-the-shelf 2D object detector and real images for training.

---

> ### Author Response · Authors · 2022-08-02
> **The response for the major concerns**
>
> We thank the reviewers for the insightful suggestions. We address the major concerns below:
> >**Q1:** The proposed method needs off-the-shelf 2D object detector.
>
> **R1:** In practice, the need for an off-the-shelf 2D object detector can be eliminated by leveraging 2D-2D feature matching. This issue has been addressed in Section 2 in the supplementary material of OnePose[30] (https://zju3dv.github.io/onepose/files/onepose_supp.pdf).
>
> Following OnePose, We first perform multiple 2D-2D feature matching between reference-query image pairs, and then select the image pair with the most inliers to estimate 2D affine transformation. The region of interest(RoI) in the query image is then detected by transforming the corner of RoI in the reference image with the estimated transformation.
>
> To validate the effectiveness of this method, we present the evaluation results on the OnePose dataset with the feature-matching-based 2D detector below. The results demonstrate that the performance of the proposed method does not degrade significantly.
>
> |   |  1cm1deg   |  3cm3deg   |   5cm5deg    |
> |:- |:-   |:-   |:-     |
> | Ours use GT bounding box (reported in the paper)  |  **50.4**   |  80.0   |  87.0     |
> | Ours use bounding box from feature matching 2D detector | 49.6 | **80.4** | **87.2**|
>
> >**Q2:** The proposed method needs real images for training.
>
> **R2:** Our 2D-3D matching network is trained on real images but can be generalized to novel objects. Besides, our method needs an object video for building an SfM model, but we don't think this is a disadvantage, as in most scenarios capturing a video of an object is much easier than acquiring its CAD model or doing object-specific training.
>
> >**Q3:** CAD model or its equivalent can be reconstructed from the movie of target object with surrounded AR markers. This might decrease the advantage of the proposed method.
>
> **R3:** Thank you very much for your comments. Leveraging AR markers can only help solve the camera poses, while dense reconstruction itself requires other modules, and the quality of the dense reconstruction is not guaranteed, especially for low-textured objects. Moreover, dense reconstruction also requires more computation than our SfM-based pipeline. Therefore we believe it is not ideal for our one-shot scenario.
>
> >**Q4:** How about processing time for training and testing?
>
> **R4:** As detailed in L236-238, our 2D-3D matching network is trained on the OnePose training set. The training takes about 20 hours with a batch size of 32 on 8 NVIDIA-V100 GPUs. At test time, our matching module runs at 87ms for a 512 × 512 query image on a single V100 GPU.
>
> >**Q5:** Is the pose estimation network trained for each object?
>
> **R5:** No. As described in L28-29, our method eliminates the need for per-object training and CAD models. Therefore, it is more applicable for AR scenarios.

---

### Meta-Review · Area_Chair_roFy · 2022-08-24

**Recommendation:** Accept
**Confidence:** Less certain

**Metareview:**

This paper originally received slightly positive reviews overall, except for one review, which was plenty of requests of specific clarifications and comments. Main issues regarded just the need of clarifying some parts of the method and put better in context of the state of the art and former evaluations. Unclear novelty was another raised problem, as well as the need of off-the-shelf 2D object detector and real images for training, which might affect the general applicability of the method while weakening the "one/shot" claim of the work. A lot of concerns were also raised about missing baselines and prior work discussion.
Authors provided detailed answers to the comments, also engaging in long discussions, especially with the most critical reviewer.
In the end, the most positive reviewers seem to be satisfied of the answers to their comments, maintaining the original positive ratings, and also the critical reviewer resulted convinced of the discussion with authors, raising his/her score to weak accept.
Overall, assuming that the comments and discussions could be included in the final version, this paper can be considered acceptable for NeurIPS 2022 publication.


**Award:**

No

---

### Decision · Program_Chairs · 2022-09-14

Accept